



**Application of Wave-current coupled Sediment Transport Models with**
**Variable Grain Properties for Coastal Morphodynamics: A Case Study of the**
**Changhua River, Hainan**
**Yuxi Wu[1a], Enjin Zhao[1b, *], Xiwen Li[2c] and Shiyou Zhang[2d]**
*1.  College of Marine Science and Technology, China University of Geosciences, Wuhan 430074,*
*China.*
*2.  Haikou Marine Geological Survey Center, China Geological Survey, Haikou 570100, China*
————————————————————
* Corresponding author: Enjin Zhao (zhaoej@cug.edu.cn)
[a] Ms., Master of *China University of Geosciences, Wuhan,* Email: yuxiwu@cug.edu.cn
[b] Ph.D., Professor of *China University of Geosciences, Wuhan,* Email: zhaoej@cug.edu.cn
[c] Senior Engineer of *Haikou Marine Geological Survey Center,* Email: lxw1818168@163.com
[d] Assistant engineer of *Haikou Marine Geological Survey Center,* Email: 460305864@qq.com





## Abstract

This study presents an integrated sand transport model that accounts for both wave and
current actions, along with non-constant grain properties, to investigate sediment dynamics in the
lower reaches of rivers. Taking the downstream and estuary of the Changhua River in Hainan
Island as a case study, topographic data and sediment sampling were conducted in the field,
complemented by remote sensing techniques. The model was rigorously validated using
theoretical and empirical methods, demonstrating excellent agreement with observed suspended
sediment concentrations at the Baoqiao Station. The findings indicate significant sediment
deposition in the estuary and lower reaches of the Changhua River, influenced by a combination of
hydrodynamic conditions and geological settings. Deposition in the estuary is primarily affected
by the northeast-southwest coastal currents and wave action, while deposition in the river channel
is associated with river constriction and variations in flow velocity. The models and methods
developed in this study provide a scientific basis for sediment management and coastal evolution
in similar downstream riverine environments and discuss the feasible scheme of sediment control
in the downstream of Changhua River.
**Keywords:** Sand transport model, Wave-current interaction, Non-constant sediment properties,
Changhua River, Hainan Island

## Plain language significance statement

This study develops an integrated sand transport model to explore sediment dynamics in river
downstream, focusing on the Changhua River estuary in Hainan Island. The research is crucial as it
addresses the complex interplay between waves, currents, and sediment movement, key to
estuarine ecosystems and shoreline changes. Our model, verified with field data, reveals
significant sediment deposition patterns influenced by coastal currents and geological features.



The findings are vital for coastal management, offering insights into how sedimentation can be
monitored and controlled. This work suggests that similar models could be applied to other river
systems, potentially guiding sustainable coastal development and protection strategies.

## 1. Introduction

Hainan Island has an extensive coastline, making marine economy a crucial source of its
economic prosperity (Feng et al., 2021, Jin et al., 2008, Fang et al., 2021). Changhua River is the
second largest river in Hainan in terms of its basin area (Zhang et al., 2020, Zeng and Zeng, 1989),
which flowing uniquely into the Beibu Gulf in the northwest of Hainan Island, serves as a crucial
water source for the region, supporting irrigation, power generation, and water supply (Yang et al.,
2013, Wang et al., 2023). The Changhua River is divided into upper, middle, and lower reaches
based on its natural geographical characteristics: the upper reaches extend from the source to
Poyang with a length of 79 kilometers and an average gradient of 14.87 %; the middle reaches run
from Poyang to Chahe with a total length of 84 kilometers, which includes a significant drop at
Guangba in Dongfang County, and generally feature a milder gradient; the lower reaches start
from Chahe down to the river's mouth at Changhua Port, spanning 39 kilometers with an average
gradient of 0.41 %, leading to a broad river plain (Figure 1). Characterized by a gentler gradient
and slower flow, the lower reaches are where the river's capacity to carry sediment decreases,
leading to increased sediment deposition. Currently, the issues related to water and sediment in the
lower reaches of Changhua River are primarily divided into studies on sediment composition and
sediment transport (Zhang et al., 2006, Wu et al., 2012, Zhu et al., 2020, Gao et al., 2014, Wang et
al., 2022, Zhao et al., 2021). About the sediment concentration information, the annual sediment
concentration of the Changhua River is recorded as 0.173 kg/m$^3$, with an average annual





sediment discharge of 782,000 tons, classifying it as a river with relatively low sediment load.
From 2013 to 2021, the average sediment concentration at Baoqiao Station in the lower reaches of
the Changhua River was determined to be 0.1227143 kg/m$^3$.

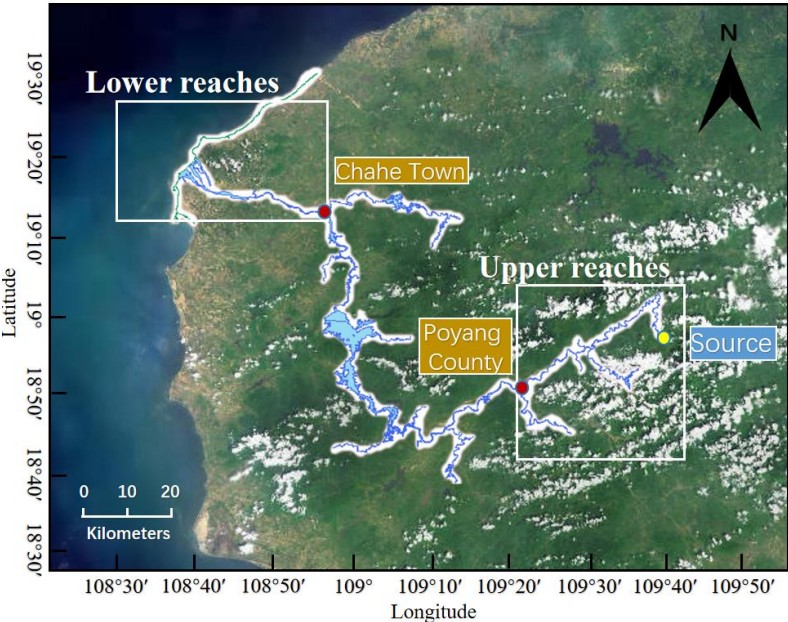


Figure 1 Division of the Upper, Middle, and Lower Reaches of the Changhua River (map origination:
https://hainan.tianditu.gov.cn/)

In the lower reaches of rivers, sediment dynamics are influenced by both water flow and

waves, which are crucial for understanding the changes in estuarine and nearshore ecosystems,
shoreline evolution, and the development of ocean resources. With the rapid advancement of
computational technologies, significant progress has been made in sediment modeling studies,
particularly in modeling sediment transport in the lower reaches of rivers where wave and current
interactions are considered.

Researchers have developed a variety of computational models to simulate sediment

transport processes in the lower reaches. These models include one-dimensional (1D),





two-dimensional (2D), and three-dimensional (3D) hydrodynamic and sediment transport models
that describe the flow and sediment movement in rivers, lakes, and coastal areas (Papanicolaou et
al., 2010). 1D models are typically used for large-scale, long-term sediment transport issues
(Thomas and Prashum, 1977, Holly and Rahuel, 1990, Papanicolaou et al., 2004), while 2D and
3D models are more suitable for simulating specific flow and sediment transport conditions,
especially in the lower reaches and estuary areas (Lee et al., 1997, Jia and Wang, 1999, Gessler et
al., 1999, Wu et al., 2000, Blumberg and Mellor, 1987).

Traditional sediment transport models have predominantly focused on the dynamics of water

flow, with wave action often addressed in a simplified manner or neglected altogether (Bakhtyar et
al., 2009, Lee et al., 1997, Spasojevic and Holly, 1990, Bai et al., 2017). We need more accurate
and comprehensive models that can describe and predict sediment behavior under the combined
action of waves and currents, especially for rivers with low sediment concentration. In this context,
the Van Rijn formula emerges as a critical tool for enhancing the precision of sediment transport
modeling (Van-Rijn, 1984). Originally formulated to calculate the transport of bed load and
suspended sediment, the Van Rijn formula has been adapted over time to accommodate the
intricate interplay between waves and currents. Its empirical nature, grounded in extensive field
and laboratory data, allows for a nuanced representation of sediment dynamics in coastal
environments. The recent applications of the Van Rijn formula in computational models have
further expanded its utility, providing a robust framework for analyzing sediment behavior in
scenarios characterized by wave and current interactions (Chen et al., 2024, Michel et al., 2023,
Addison – Atkinson et al., 2024).

With the advancement of computational technologies and the development of remote sensing



techniques, researchers have begun to incorporate the complex interactions of waves and currents
into sediment transport modeling (Han et al., 2022, Liu et al., 2014, Vinzon et al., 2023). These
models not only consider the velocity and direction of water flow but also account for the energy
input from waves, wave form changes, and the shear forces generated by wave-current interactions.
Studies have shown that sediment movement under wave action is not only influenced by the shear
stress of the water flow but also by the liquefaction and mass transport of bottom sediment caused
by waves (Niu et al., 2023). Additionally, the physical properties of sediment, such as particle size
distribution, concentration, and sedimentation rates, are crucial factors affecting sediment
behavior under the combined influence of waves and currents (Constant et al., 2023, Salgado
Terêncio et al., 2023).
Despite the progress made, sediment modeling under the combined action of waves and
currents still faces many challenges. For example, how to better simulate sediment transport in
complex turbulent flows, the coupling of flow and sediment transport, and the transport of
non-uniform sediment still require further research. Moreover, model input and calibration also
require more field data and experimental validation to ensure the reliability and applicability of the
models. To verify the effectiveness of wave-current coupled sediment model in rivers with low
sediment concentration, we take Changhua River in Hainan Province as an example to verify it.
To sum up, the sediment simulation considering only water flow can no longer meet the
accuracy of sediment prediction, and there are still limitations in the verification of sediment
simulation considering the interaction of waves and water flow. Most river sediment models do not
study rivers with small sediment concentration separately and lack in-situ observation, so the
accuracy of the models needs further verification. Additionally, due to the small scope of the lower





reaches of Changhua River, the existing terrain extraction methods are not enough to provide
terrain data with appropriate accuracy. Moreover, the sediment concentration of Changhua River is
not large and the existing research data are limited. In the absence of topographic data and
sediment data, a complete and mature sediment transport model has not been established in the
lower reaches of Changhua River so far. In this paper, we take Changhua River in Hainan Province
as a representative of the river with less sediment, and consider the sediment deposition under the
combined action of waves and currents. Based on the measured topographic data and sediment
sampling data, the bed load and suspended sediment load are calculated respectively by Van Rijn
model, and the sediment model is established. The sediment transport rate method and in-situ
observation of suspended sediment concentration are used to verify the model and analyze the
sediment deposition in the lower reaches channel and estuary.

## 2. Research Methods

## 2.1 Combined Wave and Current Sand Transport Model

The ocean hydrodynamic simulation in this study is based on the solution of the three-
dimensional incompressible Reynolds-averaged Navier-Stokes equations, with adherence to the
Boussinesq and hydrostatic pressure assumptions, namely the shallow water equations. The
specific governing equations are as follows:

$$\frac{\partial h}{\partial t} + \frac{\partial(h\bar{u})}{\partial x} + \frac{\partial(h\bar{v})}{\partial y} = hS \tag{1}$$

$$\frac{\partial h\bar{u}}{\partial t} + \frac{\partial h\bar{u}^2}{\partial x} + \frac{\partial h\overline{vu}}{\partial y} = -f\bar{v}h - gh\frac{\partial\eta}{\partial x} - \frac{h}{\rho_0}\frac{\partial p_a}{\partial x} - \frac{gh^2}{2\rho_0}\frac{\partial\rho}{\partial x} + \frac{\tau_{sx}}{\rho_0} - \frac{\tau_{bx}}{\rho_0} - \frac{1}{\rho_0}(\frac{\partial S_{xx}}{\partial x} + \frac{\partial S_{xy}}{\partial y}) + \frac{\partial}{\partial x}(hT_{xx}) + \frac{\partial}{\partial y}(hT_{xy}) + hu_s S \tag{2}$$



$$\frac{\partial h\overline{\upsilon}}{\partial t}+\frac{\partial h\overline{u\upsilon}}{\partial x}+\frac{\partial h\overline{\upsilon}^2}{\partial y}=f\overline{u}h-gh\frac{\partial \eta}{\partial y}-\frac{h}{\rho_0}\frac{\partial p_a}{\partial y}-\frac{gh^2}{2\rho_0}\frac{\partial \rho}{\partial y}+\frac{\tau_{sy}}{\rho_0}-\frac{\tau_{by}}{\rho_0}-\frac{1}{\rho_0}(\frac{\partial S_{yx}}{\partial x}+\frac{\partial S_{yy}}{\partial y})+\frac{\partial}{\partial x}(hT_{xy})+\frac{\partial}{\partial y}(hT_{yy})+h\upsilon_s S \ (3)$$

$$T_{xx}=2A\frac{\partial \overline{u}}{\partial x}, \ T_{xy}=A\left(\frac{\partial \overline{u}}{\partial y}+\frac{\partial \overline{\upsilon}}{\partial x}\right), \ T_{yy}=2A\frac{\partial \overline{\upsilon}}{\partial y} \tag{4}$$

Where $t$ is time; $x$ and $y$ are Cartesian coordinates; $\eta$ is water level; $d$ is static water
depth; $h$ is total water depth ($h=\eta+d$); $u$ and $\upsilon$ are velocity components in the $x$ and $y$
directions, respectively; $f$ is Coriolis coefficent, where $f$ represents the latitude and denotes
the Earth's angular rotation speed; $g$ is acceleration due to gravity; $\rho$ is density of water; $\tau$ is
components of radiative stress; $S$ is source-sink term; $S_{xy},S_{xx},S_{yx},S_{yy}$ are components of the
radiation stress tensor; $T_{ij}$ is the lateral stresses include viscous friction, turbulent friction and
differential advection.
This study assumes the sediment to be non-viscous, and the sediment deposition model
utilizes the results from the hydrodynamic model as open boundary driving forces. The model
definition in the sand transport model is assumed as combined current and waves, calculating the
bed load and suspended load separately. Bed load typically occurs close to the bed, while
suspended load can be transported at various levels within the water column. Sediment particles
begin to move and may become suspended when the bed shear stress exerted by waves and
currents exceeds a critical threshold. The equations adopt Van Rijn model. Van Rijn proposed the
following models for sediment transport of bed load and suspended load, which are suitable for
sediment transport calculation under wave action (Van Rijn, 1984). The Van Rijn model formula is
derived based on a set of variables that are crucial for understanding sediment transport dynamics,
particularly in the context of rivers and coastal waters. These variables include:

$$q_s=f_{sl}\cdot C_a\cdot u_*^2 \tag{5}$$





$$q_b = 0.053 \frac{M^{2.1}}{D_*^{0.3}} \sqrt{(s-1) g \cdot d_{50}^3} \tag{6}$$

$$f_{sl} = C' \cdot \left( \frac{u_*}{u_s} \right)^m \tag{7}$$

$$u_* = \sqrt{\frac{\tau}{\rho}} \tag{8}$$

$$M = \left( \frac{u_{f'}}{u_{f,c}} \right) - 1 \tag{9}$$

$$u_{f,c} = \sqrt{\theta_c (s-1) g \cdot d_{50}} \tag{10}$$

$$u_{f'} = V \frac{\sqrt{g}}{C'} \tag{11}$$

$$C' = 18 \log \left( \frac{4h}{d_{50}} \right) \tag{12}$$

$$D_* = d_{50} \sqrt[3]{\frac{(s-1) g}{v^2}} \tag{13}$$

Where $q_b$ is the bed load transport rate; $q_s$ is the suspended load transport rate; $M$ is
the non-dimensional transport stage parameter; $u_{f,c}$ is the critical friction velocity, which under
the current; $\theta_c$ is the critical Shield parameter; $u_{f'}$ is the effective friction velocity; $C'$ is the
Chezy number originationg from skin friction; $D_*$ is the non-dimensional particle parameter; $v$
is the kinematic viscosity and approximately equal to $10^{-6}$ m$^2$/s for water; $C_a$ is the bed
concentration; $u_*$ is the friction velocity; $\tau$ is the shear stress at the bed surface; $\rho$ is the
density of water; $m$ is empirical exponent.
In the context of the Van Rijn model, the non-dimensional particle parameters can influence





the value of the critical Shield parameter. For example, as the particle size increases, the critical
Shields parameter may also increase because larger particles require more force to overcome
gravity and initiate motion. Similarly, changes in fluid properties or flow conditions can affect both
the non-dimensional particle parameters and the critical Shield parameter. Instead of using a
constant critical Shields parameter $\theta_c$ , Van Rijn assumes the following variation as a function of
$D_*$ , see Figure 2.

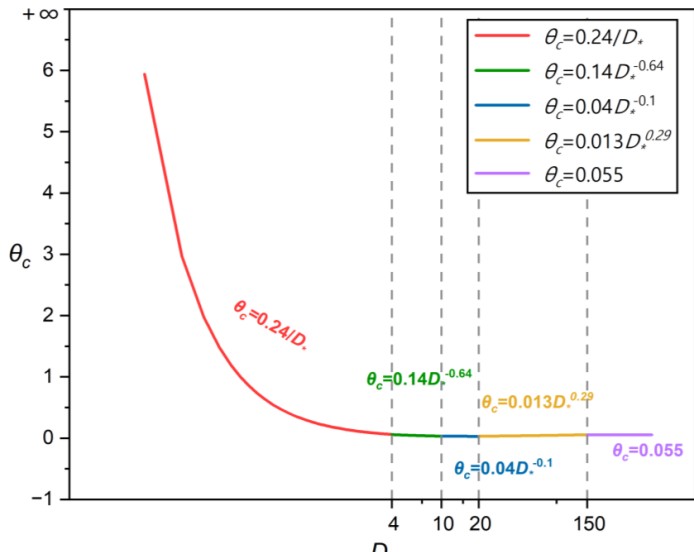

Figure 2 Relations for determination of critical Shields stress

After calculating the bed load and suspended load separately, the Bijker model is used to

calculate the total sediment transport rate (Bijker, E.W. 1967), which includes both bed load and
suspended load components. and the formula is as follows:

$$q_t = q_s + q_b = q_b \left(1 + 1.83Q\right) \tag{14}$$

$$Q = A\left(\frac{I_1}{I_2}\right) + I_2 \ln\left(\frac{z^*}{r}\right) = \frac{h}{r}\left(\frac{I_1}{I_2}\right) + I_2 \ln\left(\frac{w}{rku_{f,wc}}\right) \tag{15}$$



$$u_{f,wc} = u_{f,c} + \sqrt{u_{f,c}^2 + 2 \cdot \frac{\upsilon^2}{V}} \qquad (16)$$

$$I_1 = \int_0^h \frac{u(z)}{w} dz, \quad I_2 = \int_0^h \frac{u(z)}{w} \ln(\frac{h-z}{d_{50}}) dz, \qquad (17)$$

Where $q_t$ is the total sediment transport rate; $Q$ is a dimensionless factor that accounts
for the effect of waves on the bed load transport; $h$ represents the water depth; $r$ is the bed
roughness; $I_1$ and $I_2$ are Einstein's integrals, which are functions of the dimensionless
reference level A and the dimensionless roughness height $z^*$; $w$ is the settling velocity of the
suspended sediment; $k$ is von Karman's constant; $u_{f,wc}$ is the shear velocity under the
influence of combined waves and current; $\upsilon$ is the amplitude of the wave-induced oscillatory
velocity at the bottom; $V$ is the depth-averaged flow velocity; $u(z)$ is the flow velocity
profile at a height $z$ above the bed.

## 2.2 Influences of Waves and Currents

The influence of sediment transport model on water flow has been widely studied and
applied (Papanicolaou et al., 2010), including sediment transport mechanisms, the establishment
of the boundary layer, modifications to bed morphology, and the vertical distribution of
suspended sediment. However, the theory and application of wave action are not mature
compared with water flow. This chapter emphasizes the motion equation and boundary condition
equation adopted by wave action in the sediment transport model in this paper.
The model of sediment transport to calculate the influence of the waves usually through a
comprehensive consideration of various factors that encapsulate the impact of waves on sediment
transport. The typical models incorporate the nonlinear characteristics of wave motion, net mass



transport induced by waves, turbulence generated by wave breaking, the temporal evolution of
the boundary layer due to combined wave and current action, contributions to turbulence from
three sources (wave boundary layer, mean flow, and wave breaking), and the influence of
wave-formed ripples on flow and sediment transport. A suite of wave theories, such as Stokes
and Cnoidal theories, are employed to describe wave motion across different hydrodynamic
conditions. Additionally, the model accounts for the calculation of turbulence viscosity due to
wave breaking, and the equations to compute the shear stress resulting from wave motion are
well represented. These complex interactions and processes are articulated through a series of
mathematical equations and empirical formulas, enabling the model to accurately simulate the
process of sediment transport under the dual influence of waves and current. In this paper, the
specific formulas of the wave motion are as follows:

**Table 1 Formulas of the wave motion in the sand transport model**

| Item | Method | Equation |
|---|---|---|
| Wave Energy Dissipation | Battjes and Janssen (1978) | $D = \dfrac{\gamma_1 g H^2}{\gamma_2 k}\tanh\left(\gamma_2 kh\right)$ |
| Wave Boundary Layer Thickness | Empirical formula | $\delta = \dfrac{k}{30}\left(\dfrac{u_{max}}{u_*}\right)$ |
| Turbulent Viscosity Induced by Waves | Empirical formula | $v_t = C_\mu \dfrac{u_{max}^2}{g}$ |
| Shear stress resulting from wave motion | Jørgen Fredsøe (1984) | $\tau = \rho u_*^2$ |
| Wave velocity in shallow water | Cnoidal theory | $c = \sqrt{gh}[1 + \dfrac{H}{h}(\dfrac{1}{k^2} - 0.5 - \dfrac{3E(x)}{2k^2 K(x)})]$ ( $k$ is the module of elliptic function. $E(x)$ and $K(x)$ are the first and second complete elliptic integrals) |
| Wave velocity in deep water | Stokes theory | $c = \sqrt{\dfrac{g\lambda}{2\pi}}$ |

Additionally the influence of waves and currents on the sediment transport model, sediment

parameters are the direct conditions that affect the accuracy of the model, as follows.



## 2.3 Non-constant sediment properties

Generally speaking, sediment data may have different particle size, sorting, porosity and relative density equivalence, and are not uniform. These characteristics lead to the increase of computational complexity (Adnan et al., 2019), so most of studies set the sediment parameters in the study area as a constant parameter for calculation (Mohd Salleh et al., 2024, Auguste et al., 2021). Actually, the spatial distribution of sediment parameters is not constant. Seabed sediment is not homogeneous, and as the distance from the shore increases, the grain size of the deposited sediment continuously decreases. Some researches had proved the validity of sand transport model with spatially variable sediment properties (Doroudi and Sharafati, 2024, Bui and Bui, 2020). Sediment properties can be obtained by direct method and indirect method. The indirect method includes theoretical formula and empirical formula, while the direct method is sampling (Claude et al., 2012, Leary and Buscombe, 2020). Studies had shown that indirect methods are less effective than direct sampling (Claude et al., 2012). In this paper, sediment sampling is conducted using a clam grab sampler to collect surface geological samples from targeted sea areas. The study area is divided into river channel and estuary segments, with sediment samples collected at consistent intervals (Figure 3). We sampled 15 points in estuary of the lower reaches of Changhua River and 40 points in the riverway. To ascertain sediment parameters, including grain size and sorting factors, a laser particle size analyzer is utilized.



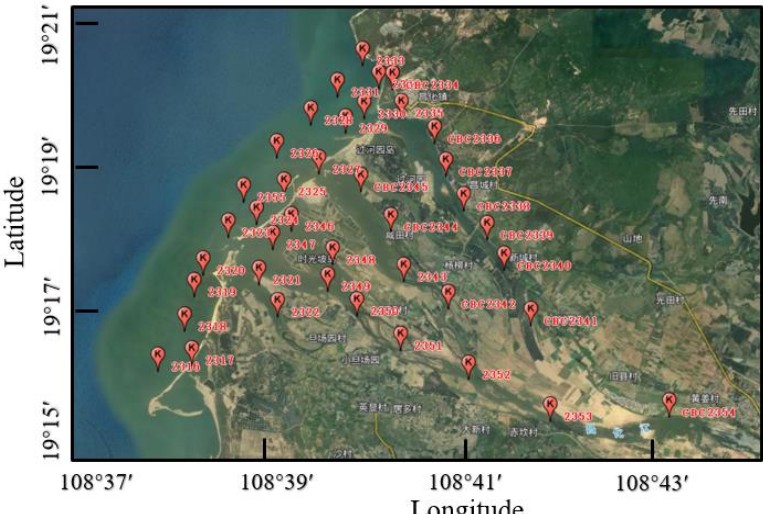


Figure 3 Location of sediment sampling point (map origination: https://hainan.tianditu.gov.cn/)

After selection, the analytical process detailed particle size and sediment segregation data
(Table 2 and Table 3). Grain size parameters are quantitative representations of the grain size
characteristics of the clastic material in terms of certain values. The individual grain size
parameters and their combined characteristics can be used as the basis for discriminating the
depositional hydrodynamic conditions and depositional environment. The commonly used
parameters are mean particle diameter (Mz), sorting coefficient ($\delta_i$) and median grain diameter
($\Phi_{50}$). The number of samples at the estuary with a median grain diameter between 0 and $1\varphi$ is
9, accounting for 60 %; the number of samples with a mean grain size between $1\varphi$ and $3\varphi$ is 3,
accounting for 20 %; the number of samples with a median grain size between $-1\varphi$ and 0 is 3,
accounting for 20 %. While, in the estuary and 40 points in the lower reaches, the number of
samples with a median grain diameter between 0 and $1\varphi$ is 24, accounting for 60 %; the number
of samples with a median grain size between $1\varphi$ and $3\varphi$ is 8, accounting for 20 %; the number
of samples with a median grain size between $3\varphi$ and $7\varphi$ is 7, accounting for 17.5 %; the



number of samples with a median grain size between -1φ  and 0 is 1, accounting for 2.5 %.

**Table 2 Grain parameters of samples at the estuary**

|  | Coefficient of granularity | | | |
| :---: | :---: | :---: | :---: | :---: |
| Number | Mean grain diameter Mz(φ) | Sorting factor δi(φ) | Median grain diameter Φ50(φ) | Classification of sediments |
| 1 | <0.04 | 0.7600 | 0.02 | Gravel sand |
| 2 | <0.04 | 1.1000 | -0.44 | Sandy gravel |
| 3 | 0.33 | 0.7600 | 0.33 | Sand |
| 4 | <0.04 | 0.7900 | 0.01 | Silty sand |
| 5 | 0.50 | 0.7700 | 0.51 | Sand |
| 6 | 0.40 | 0.8200 | 0.41 | Sand |
| 7 | 0.98 | 0.6500 | 1.00 | Sand |
| 8 | 1.35 | 0.6900 | 1.41 | Sand |
| 9 | 2.91 | 0.9600 | 2.87 | Sand |
| 10 | 0.31 | 0.7700 | 0.32 | Sand |
| 11 | 0.26 | 0.7600 | 0.27 | Sand |
| 12 | <0.04 | 0.6700 | -0.41 | Sandy gravel |
| 13 | <0.04 | 0.8000 | -0.15 | Silty sand |
| 14 | 0.18 | 0.7700 | 0.19 | Sand |
| 15 | 0.70 | 1.2900 | 0.69 | Sandy gravel |


**Table 3 Grain parameters of samples of the river**

|  | Content of grain(%) | | | | Coefficient of granularity | | | |
| :---: | :---: | :---: | :---: | :---: | :---: | :---: | :---: | :---: |
| Number | Gravel | Sand | Silt | Clay | Mean grain diameter Mz(φ) | Sorting factor δi(φ) | Median grain diameter Φ50(φ) | Classification of sediments |
| 1 | 0.00 | 8.55 | 83.90 | 7.55 | 6.01 | 1.42 | 6.09 | Silt |



| | | | | | | | | |
|---|---|---|---|---|---|---|---|---|
| 2 | 0.00 | 70.64 | 26.48 | 2.88 | 3.33 | 2.14 | 2.44 | Silty sand |
| 3 | 0.00 | 85.98 | 13.06 | 0.96 | 2.82 | 1.26 | 2.79 | Silty sand |
| 4 | 0.00 | 87.44 | 6.38 | 0.45 | 2.64 | 1.27 | 2.57 | Silty sand |
| 5 | 5.90 | 93.12 | 0.98 | 0.00 | 0.15 | 0.75 | 0.16 | Gravel sand |
| 6 | 0.00 | 2.48 | 89.78 | 7.74 | 6.20 | 1.26 | 6.22 | Silt |
| 7 | 0.00 | 9.12 | 81.51 | 9.37 | 6.25 | 1.47 | 6.45 | Silt |
| 8 | 10.96 | 87.75 | 0.97 | 0.07 | <0.04 | 0.70 | -0.17 | Gravel sand |
| 9 | 1.18 | 98.02 | 0.75 | 0.05 | 0.51 | 0.72 | 0.52 | Gravelly sand |
| 10 | 8.18 | 90.50 | 1.21 | 0.11 | 0.12 | 0.84 | 0.11 | Gravel sand |
| 11 | 4.42 | 92.40 | 2.95 | 0.23 | 0.30 | 0.83 | 0.29 | Gravelly sand |
| 12 | 3.56 | 91.40 | 4.77 | 0.46 | 0.79 | 1.33 | 0.74 | Gravelly sand |
| 13 | 0.03 | 96.04 | 3.57 | 0.36 | 1.17 | 0.86 | 1.17 | Gravelly sand |
| 14 | 1.13 | 91.58 | 6.84 | 0.45 | 1.24 | 1.40 | 1.16 | Gravelly sand |
| 15 | 1.51 | 95.25 | 2.90 | 0.33 | 0.71 | 0.92 | 0.68 | Gravelly sand |
| 16 | 0.00 | 94.96 | 4.68 | 0.35 | 1.32 | 1.00 | 1.31 | Sand |
| 17 | 0.00 | 96.21 | 3.47 | 0.32 | 1.34 | 0.81 | 1.33 | Sand |
| 18 | 0.00 | 98.26 | 1.40 | 0.34 | 1.21 | 0.71 | 1.20 | Sand |
| 19 | 0.00 | 17.37 | 74.44 | 8.20 | 5.89 | 1.81 | 6.33 | Sandy silt |
| 20 | 0.00 | 1.61 | 89.02 | 9.37 | 6.33 | 1.27 | 6.39 | Silt |
| 21 | 4.70 | 47.88 | 42.65 | 4.52 | 3.43 | 3.20 | 3.69 | Gravelly muddy sand |
| 22 | 28.43 | 71.40 | 0.12 | 0.05 | 0.69 | 0.84 | 0.75 | Gravel sand |
| 23 | 4.01 | 45.93 | 44.98 | 5.07 | 3.57 | 3.19 | 3.99 | Gravelly mud |
| 24 | 3.26 | 75.71 | 20.00 | 1.42 | 1.77 | 2.53 | 0.63 | Gravelly muddy sand |
| 25 | 0.05 | 98.99 | 0.88 | 0.08 | 0.91 | 0.70 | 0.92 | Gravelly sand |
| 26 | 2.86 | 91.07 | 5.73 | 0.34 | 0.67 | 1.29 | 0.62 | Gravelly sand |
| 27 | 40.14 | 60.58 | 14.52 | 1.16 | 1.54 | 2.66 | 0.39 | Muddy sandy gravel |
| 28 | 24.57 | 69.98 | 4.97 | 0.47 | 0.13 | 1.43 | 0.11 | Gravel sand |



| 29 | 26.79 | 69.74 | 3.26 | 0.21 | 0.56 | 0.99 | 0.55 | Gravel sand |
| 30 | 36.45 | 72.08 | 4.72 | 0.40 | 0.21 | 1.35 | 0.22 | Sandy gravel |
| 31 | 5.23 | 92.23 | 2.34 | 0.20 | 0.30 | 0.83 | 0.30 | Gravel sand |
| 32 | 0.79 | 99.21 | 0.00 | 0.00 | 0.73 | 0.75 | 0.75 | Gravelly sand |
| 33 | 4.06 | 95.54 | 0.68 | 0.08 | 0.44 | 0.82 | 0.47 | Gravelly sand |
| 34 | 17.53 | 73.84 | 8.00 | 0.63 | 0.36 | 1.59 | 0.29 | Gravelly muddy sand |
| 35 | 0.85 | 99.15 | 0.00 | 0.00 | 0.64 | 0.72 | 0.65 | Gravelly sand |
| 36 | 38.74 | 67.26 | 9.86 | 0.98 | 0.58 | 1.82 | 0.33 | Muddy sandy gravel |
| 37 | 32.10 | 51.01 | 15.89 | 1.01 | 1.61 | 2.73 | 0.26 | Muddy sandy gravel |
| 38 | 52.91 | 34.33 | 11.76 | 1.01 | 1.64 | 2.91 | 0.07 | Muddy sandy gravel |
| 39 | 7.23 | 72.16 | 19.53 | 1.07 | 1.46 | 2.53 | 0.38 | Gravelly muddy sand |
| 40 | 3.81 | 90.24 | 4.21 | 0.37 | 0.38 | 0.94 | 0.45 | Gravelly sand |

The surface sediment particles in the nearshore area of Changhua river course are mainly
divided into three grain size components, gravel (>2 mm), sand (2~0.063 mm), silt (0.063~0.004
mm), with relative percentages of 9.28%, 72.18% and 18.54%, respectively. Based on the
sampling and testing results of the river course, we can obtain the histogram of the component
percentage for each sample (Figure 4). It is obvious that the sediment composition in the river
channel is dominated by sand, followed by silt.



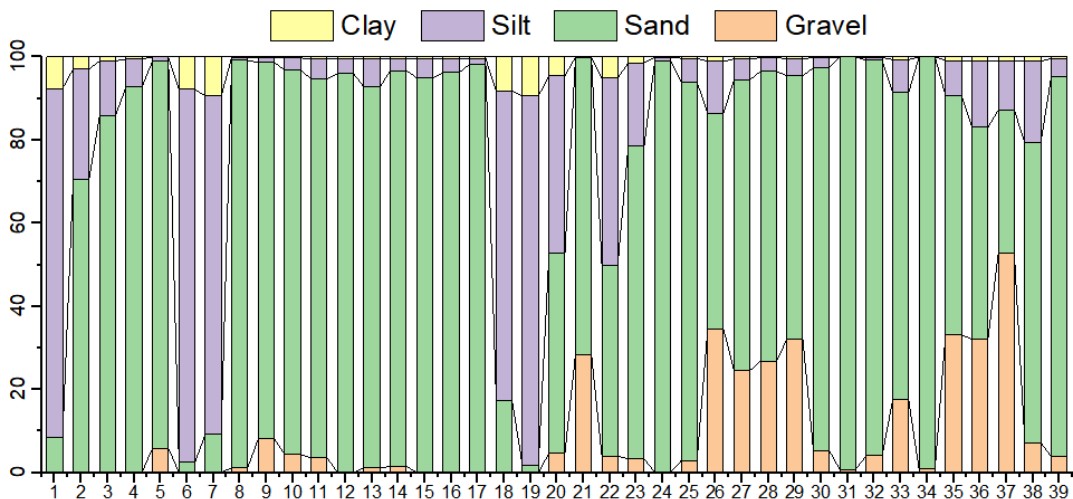

Figure 4 Percentage composition of components in river samples

According to the classification criteria of the sorting coefficients by Focke–Ward (Table 4), Sediments from the Changhua River estuary in the lower reaches exhibit medium sorting with coefficients of most samples between 0.71~1.00 and a median grain diameter predominantly under 1.5 mm, characterized mainly by sand. In contrast, sediments within the river stretch between Baoqiao Station and the lower reaches are coarser with poorer sorting, evidenced by a sorting coefficient exceeding 1.00 in 23 out of 40 samples (over 57 %).

Table 4 Sorting level table

| Sorting Grade | Sorting factor ($\delta i(\varphi)$) |
| --- | --- |
| Sorting excellent | <0.35 |
| Sorting good | 0.35~0.71 |
| Sorting medium | 0.71~1.00 |
| Sorting poor | 1.00~4.00 |

To ascertain the sediment composition and the dry bulk density in the estuary, 15 samples were collected from the Changhua River estuary. These samples were dried to measure mass and volume, thereby determining the dry bulk density of the sediment. After calculating, the dry bulk



density is 1210.9 kg/m³ which uesd in sand transport model. This analysis is crucial for model
accuracy and understanding sediment behavior in the estuarine environment. According to the
sampling position, the research area is divided into areas. After sorting and interpolation, the
spatial variation of sediment particle size data and sorting data in the study area are obtained
(Figure 5).

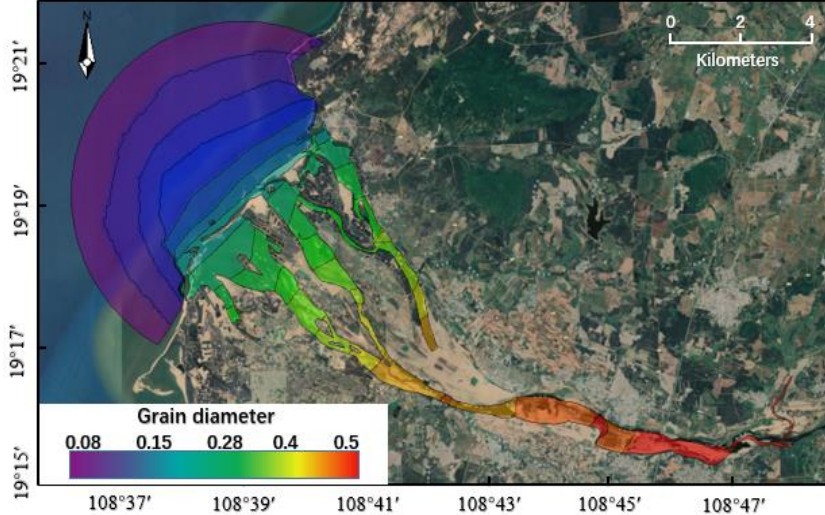


Figure 5 Two-dimensional spatial variation of sediment particle size data (map origination:
https://hainan.tianditu.gov.cn/)
**2.4 Reliability evaluation index**
In this paper, Nash-Sutcliffe model efficiency coefficient (NSE) and root mean squared
error (RMSE) are used to evaluate the reliability of the model. The calculation formulas (Nash
and Sutcliffe, 1970) are as follows:

$$NSE = 1 - \frac{\sum_{i=1}^{N}(M_i - O_i)^2}{\sum_{i=1}^{N}(O_i - \bar{O})^2} \tag{17}$$



$$RMSE = \sqrt{\frac{\sum_{i=1}^{N}(M_i - O_i)^2}{N}} \qquad (18)$$

In Equations: $M_i$ is the model simulation value at the $i$ moment; $O_i$ is the measured value at
the $i$ moment; $\overline{O}$ is the average of the measured values of the site at all simulation moments; $N$
is the total number of all simulation moments. Among them, the value range of NSE is 0~1.
When 0.65≤NSE<1, the fitting degree of the model is excellent; When 0.5≤NSE<0.65, the
fitting degree of the model is good; When 0.2≤NSE<0.5, the fitting degree of the model is
general; When 0<NSE<0.2, the fitting degree of the model is poor.
**3. Example in the lower reaches of the Changhua River**
**3.1 Model Region**
The study area is situated in the western part of Hainan Island, mainly encompassing the
lower reaches of Changhua River and its estuary. The approximate coordinates range from
108°36'E to 108°50'E and 19°15'N to 19°22'N. The study area covers a large part of the region
from Chahe Town to the estuary of the Changhua River, including towns such as Changhua
Town, Sigeng Town, Sanjia Town, and Wulie Town, among others.



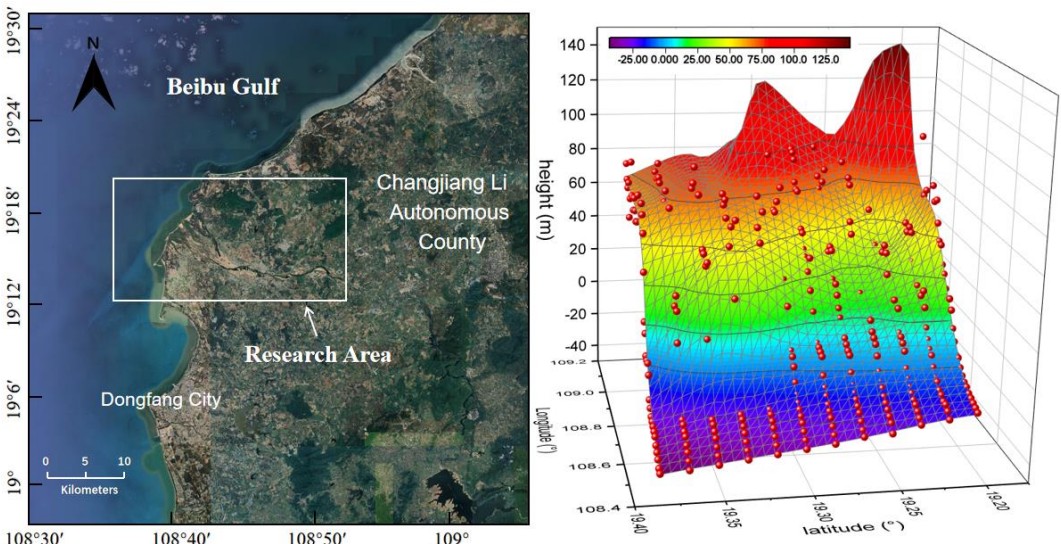

Figure 6 Scope of study area (The right figure shows the 3D terrain after the interpolation of ETOPO1 topographic data, the red dots are the original data.) (map origination: https://hainan.tianditu.gov.cn/)

In the study, bathymetric data is derived from ETOPO1 global seafloor topography data and in-situ measurements using ADCP. The spatial resolution of ETOPO1 data is $1/60°\times1/60°$, which is insufficient for the research requirements. ADCP depth measurements have higher density in nearshore areas and provide actual measured data with higher accuracy.

The model's open boundary conditions are defined by the forced tidal water level, incorporating eight primary tidal components: M2, S2, K1, O1, N2, K2, P1, and Q1. The model's closed boundary aligns with the terrestrial boundary, where the normal velocity of ocean currents is set to zero, precluding any exchange of temperature and salt between land and seawater. The time resolution of tidal level data is 1 hour and the accuracy is 1 cm. There are 121 open boundary control points. For the setup of wave conditions, this paper selects the JONSWAP spectrum for the initial condition spectrum of the boundary. The wave parameters at the open boundary are set to fixed values, referring to the annual average frequency of occurrence of wave





heights in various directions at the Dongfang Ocean Station over the years, as well as the number
of days and frequency of occurrence in different seasons for each wave level (Ding, 1990, Hu,
2009, Wang, 2023). The wave field are driven by wind, with reference to the 10-meter wind
speed and pressure parameters from the ERA5 reanalysis data provided by European Centre for
Medium-Range Weather Forecasts (ECMWF). The model also integrates the impact of wind
fields, with data sourced from ECMWF at a resolution of $1/8° \times 1/8°$. This dataset encompasses
the u (east-west) and v (north-south) components of the wind vector, along with sea level
pressure. After introducing these environmental conditions into the model, a hydrodynamic
model containing water level and flow information can be obtained. The upper boundary of the
model is set based on the multi-year average monthly flow and sediment concentration data from
the Baoqiao Hydrological Station in Chahe Town.
**3.2 Verification of hydrodynamic model**

In order to ensure the validity of the model, the tidal current data of one tide gauge station

and two ADCP points in the study area are compared and verified. Figure 7 shows the hourly
water level comparison between the measured tidal water level at Basuo Port Station (19°06'N,
108°37'E) and the model simulation results. Model validation occurs from 10:00 on April 23,
2023, to 00:00 on April 30, 2023. After calculation, the RMSE of the simulation results is 18.101
cm and the NSE is 0.9501, which is within the acceptable range. This shows that the model is
reliable and meets the demand, and can be used to simulate the tidal current in the research area
of the lower reaches of Changhua River.





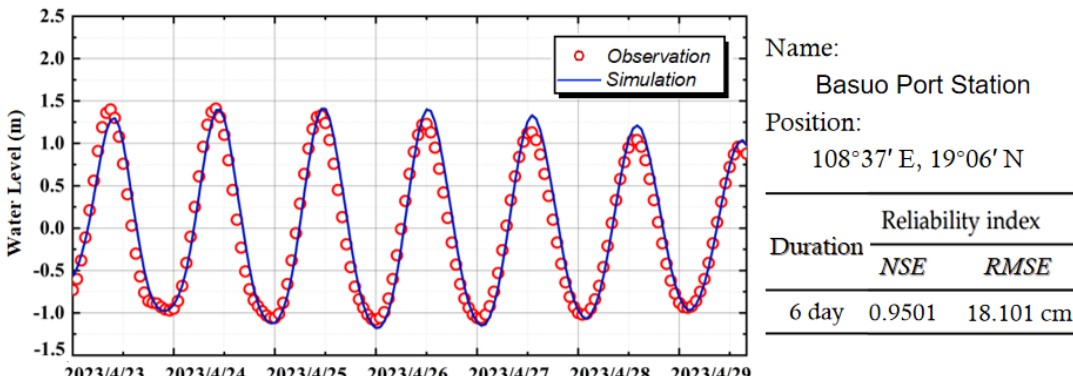

Figure 7 Hourly water level verification of Basuo Port Station

During the sea trial, two points were selected to continuously observe the velocity and direction of seawater. In order to obtain the seawater situation in lunar day, the continuous measurement time of each point was 25 hours. Information about the position and observation time of the measuring point is as follows.

Table 5 Information of fixed-point current station

| Number | Position | Observation |
|---|---|---|
| ADCP 01 | 108°37′E, 19°17′N | April 23rd at 10: 00 - April 24th at 11: 00 |
| ADCP 02 | 108°39′E, 19°20′N | April 24th at 17: 00 - April 25th at 18: 00 |

Current velocity and direction verification at the Changhua River estuary involves a 5-minute time resolution analysis using an Acoustic Doppler Current Profiler (ADCP) 01. Located over 2 km offshore with a water depth of 20.9 m, ADCP 01's data is compared against simulations at five-minute intervals. The 25-hour observation period, from 10:00 on April 23, 2023, to 11:00 on April 24, 2023, encompasses a full lunar day, providing a comprehensive dataset.

The model's simulated velocity and direction are found to be in substantial agreement with the ADCP 01 measurements, particularly in regions where tidal currents are predominant. The model accurately replicates the velocity fluctuations, affirming its capability to capture the

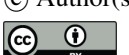

dynamics of the study area. At ADCP 01, the model's predictions are notably accurate due to the
shallow water depth and the distance from the shore, which intensify the tidal effects and make
the influence of other factors more pronounced. This results in a reduced error, validating the
model's performance. The consistency between the model and the measurements confirms the
high reliability of the model for future research applications.

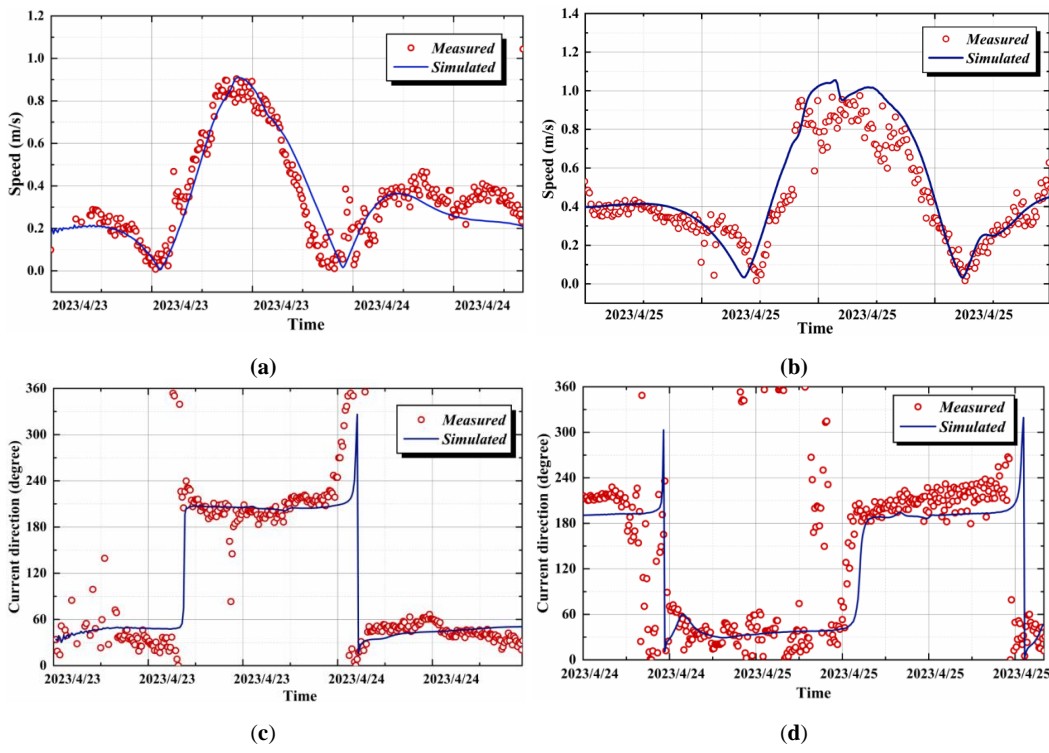

Figure 8 Current velocity and direction verification: (a) velocity verification of ADCP 01; (b) velocity
verification of ADCP 02; (c) verification of current direction of ADCP 01; (d) verification of current
direction of ADCP 02
**3.3 Results of hydrodynamic model**
The hydrodynamic simulation outcomes, as depicted in Figure 9, indicate a predominantly
NE-SW reciprocating current pattern within the study area. This flow is aligned parallel to the
coastline, with the tidal current shifting direction according to the tidal phase. During high tide,



the current is directed towards the northeast, as illustrated in Figure 9b. Conversely, during low
tide, the flow reverses, moving towards the southwest, as shown in Figure 9c. These findings are
crucial for understanding the tidal dynamics of the region.

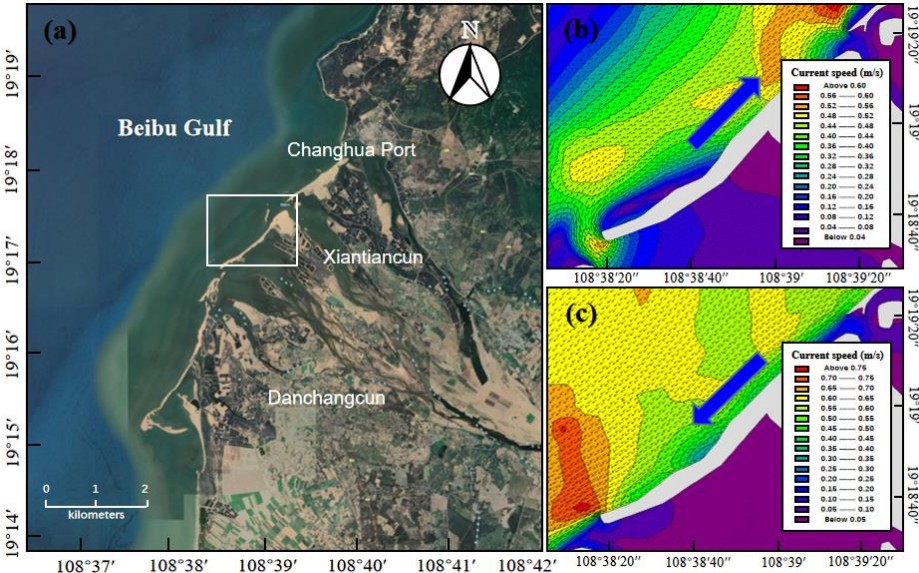


Figure 9 Study area and coastal current direction: (a) location map of the study area; (b) detailed zoom of
the map in Fig. 9a with NE current; (c) detailed zoom of the map in Fig. 9a with SW current. (map
origination: https://hainan.tianditu.gov.cn/)

### 3.4 Verification of sediment model

To validate the effectiveness of the sediment model, a combination of theoretical and
empirical validation methods is employed to verify the simulation results. Theoretical validation
is conducted using the sediment transport rate method to calculate the annual sediment
deposition thickness, and the model's effectiveness is verified by comparing the theoretical
sediment deposition thickness with the simulated changes in riverbed thickness. Empirical
validation involves comparing the measured daily suspended sediment concentration (SSC) data
from Baoqiao Station in the lower reaches of the Changhua River with the simulated values for



comparative analysis.

### 3.4.1 Theoretical validation by sediment transport rate method

After adding data such as sediment motion equation and particle size sorting to the original
hydrodynamic model, a sediment transport model under the combined action of waves and
currents is formed (Figure 10). Figure 10 shows the variation of sediment thickness in the study
area after one week of simulation.

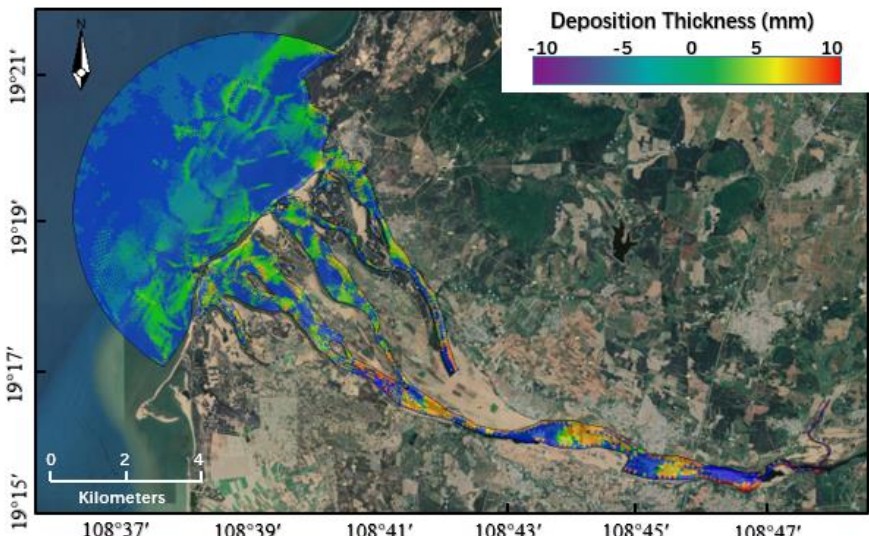

Figure 10 Sand transport result in the study area (map origination: https://hainan.tianditu.gov.cn/)
To calculate the scouring and silting volume along the river reach, the sediment transport
rate method is employed, as per the Code for Design of River Regulation. This method involves
calculating the difference in sediment mass between the upstream and downstream stations to
determine the weight of sediment scoured and deposited. This value is then divided by the
sediment's dry density to ascertain the volume of scour and deposition. The resulting data is used
to estimate the uniform scour and deposition thickness within the river reach, as outlined by the





following equation:

$$\Delta W = W_s^{upper} + W_s^{inflow} - W_s^{outflow} - W_s^{lower} \tag{19}$$

$$\Delta V = \frac{\Delta W}{\rho'} \tag{20}$$

Where: $\Delta W$ is deposition weight of the river (t); $\Delta V$ is scouring and silting volume of
river reach (m); $W_s^{upper}$ is upper station sediment quantity (t); $W_s^{inflow}$ is sediment inflow (t);
$W_s^{outflow}$ is sediment outflow (t). This usually refers to the amount of sediment diverted from the
main river channel within the river section due to some engineering water diversion or natural
water diversion (such as the confluence of tributaries).; $W_s^{lower}$ is sediment discharge at the
lower station of the river section (t). It represents the output at the end of the river section and is
the total amount of sediment passing through the downstream cross-section of the river section.;
$\rho'$ is dry density of sediment deposition (t/m$^3$).
The model calculations indicate a catchment area of 85,203.643779 km$^2$ in the lower
reaches of the Changhua River. The dry density of sediment, crucial for erosion and deposition
analysis, is determined through sediment sampling and subsequent drying, yielding an average
dry bulk density of 1.214723798 t/m$^3$ across 15 samples. Utilizing this data, the sediment
scouring and silting weight within the river channel is deduced from the estuary's sediment
discharge in 2022. Applying the formula, the estimated sediment thickness for the lower reaches
in 2022 is approximately 4.1 cm.
As depicted in Figure 11, the natural variation of sediment thickness in Changhua Port
during 2022, in the absence of human intervention, is presented. It can be seen that there is
basically no deposition from January to April, and July to August is the fastest deposition interval.
This is in line with the actual situation in the study area. The theoretical deposition thickness



assumes uniform scouring and silting distribution, which may not accurately represent areas with
significant water depth variations. The actual average silting height in Changhua Port for 2022 is
calculated to be about 5.2 cm, derived from the shallow riverbed section between Danchangcun
and Xiantiancun. This value exceeds the theoretical thickness by 1.1 cm, likely due to the
presence of a river island obstructing river flow, thereby reducing flow velocity and enhancing
sediment deposition. This discrepancy underscores the impact of local geomorphological features
on sediment dynamics.

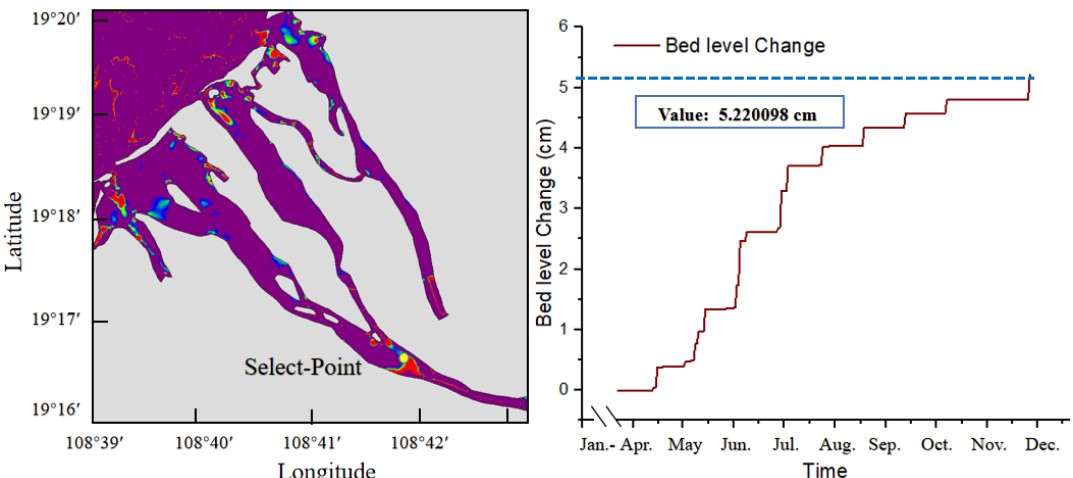


Figure 11 Selection point for sediment deposition verification

3.4.2 Empirical validation by suspended sediment concentration

In the lower reaches of the Changhua River, the summer season is the most pronounced for

sediment variation within a year, with the highest sediment concentration and sediment transport
rate (Mao et al., 2006). Therefore, sediment data from July, which is representative, are selected
for model validation. The simulated Suspended Sediment Concentration (SSC) is compared with
the daily observed SSC at Baoqiao Station for the month of July (Figure 12). The SSC at
Baoqiao Station is the highest during the first two days of July, reaching a peak SSC of 0.55



kg/m³. Subsequently, the SSC continuously decreases, reaching its lowest value on the 5th of

July, and then slowly rises. After the 10th of July, it gradually decreases from 0.301 kg/m³, with

the most values remaining below 0.2 kg/m³. Based on the analysis, NSE for Baoqiao Station is

0.8389; the RMSE is 0.097244 kg/m³. The observed SSC are in good agreement with the

simulated values.

To further analyze the simulation validation, Figure 12 presents a histogram of the daily

absolute error in SSC at Baoqiao Station. The absolute error is calculated as the absolute

difference between the measured and simulated values. The Mean Absolute Error (MAE) is

defined as the average over the test sample of the absolute differences between prediction and

actual observation. The MAE in SSC for Baoqiao Station in July is 0.071224 kg/m³. The

maximum error occurs at the beginning and the end of the month, which may be due to the use of

monthly average flow and sediment data for the model's upper boundary input, thereby

increasing the model's error. Overall, the difference between the daily observed SSC values and

the simulated results at Baoqiao Station in July is within a reasonable range, indicating that the

model has an acceptable level of precision.

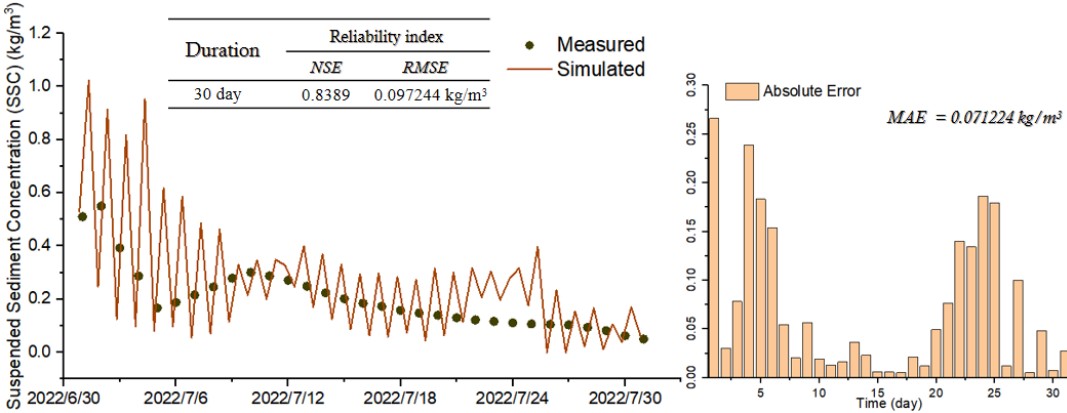

Figure 12 Selection point for sediment deposition verification



### 3.5 Analysis of depositions in Changhua River estuary

Sediment deposition in the Changhua River estuary is influenced by both hydrodynamic and geological factors. The predominant northeast-southwest coastal current direction and wave action, has led to the formation of a two-way sand mouth, further narrowing the estuary. Secondly, the estuary's geomorphology consists of a sandy riverbed with poor stability. The bed slope at the estuary decreases, and the water flow's capacity to carry sediment is reduced. Therefore, the sediment accumulation at the mouth of the Changhua River is relatively severe..

Over time, these processes have resulted in the formation of two river islands, altering the estuary into a complex channel system with multiple smaller estuaries. Currently, the main river channel flows between these islands, exhibiting shallow depths during low tide. These findings are pivotal for understanding the estuary's morphological evolution and inform strategies for sediment management in such dynamic environments.

The result of the sediment simulation (Figure 10) shows the variation of sediment thickness in the study area after one week of simulation. There are two obvious depositions in the study area, including the estuary and the slender channel. The figure clearly shows the serious and slight areas of siltation in the study area. However, the specific sedimentary characteristic in the study area is unknown, needing further analysis. To solve this problem, we extract the bed level change data of a point in the obvious change area of river bed, and take this point as the whole area. Therefore, the sediment deposition characteristic in this area can be analyzed through the bed level change at this point.

Results of Danchangcun are shown in Figure 13, which illustrates the bed level changes and consequent sediment deposition and scouring in various parts of Danchangcun. Positive values indicate sediment deposition, while negative values denote scouring.





In the estuary of Danchangcun (Figure 13b), the bed level fluctuates above zero, signifying

net sediment deposition with a final accumulation of approximately 0.59 cm over the simulation

period.

The deposition near the river island in Danchangcun (Figure 13d) follows a cyclical pattern

over a 24-hour cycle, with an overall sediment thickness of about 0.20 cm. Initially, sediment

accumulates quickly, after which the bed level stabilizes at its peak value. A sharp decrease in

deposition rate is observed in the last two hours, with each cycle adding about 0.03 cm of

sediment.

At the front end of the sand mouth (Figure 13f), the bed level decreases by 0.39 cm,

indicating active scouring and sediment removal. The continuous negative bed level changes

suggest an increasing scouring intensity, especially pronounced on April 23 when a significant

erosion event led to a 0.18 cm drop in bed level.

Finally, Figure 13h examines sediment deposition at the sand mouth, with two distinct

locations showing similar sedimentation trends, albeit with Location 2 (near the river)

experiencing faster sedimentation. Prior to April 24-25, Location 1 (near the ocean) registered

erosion, followed by a transition to net deposition, while Location 2 showed minor erosion

before April 24. The simulation predicts final bed level changes of approximately 0.42 cm for

Location 1 and 0.60 cm for Location 2.



(a)

(b)

(c)

(d)

(e)

(f)



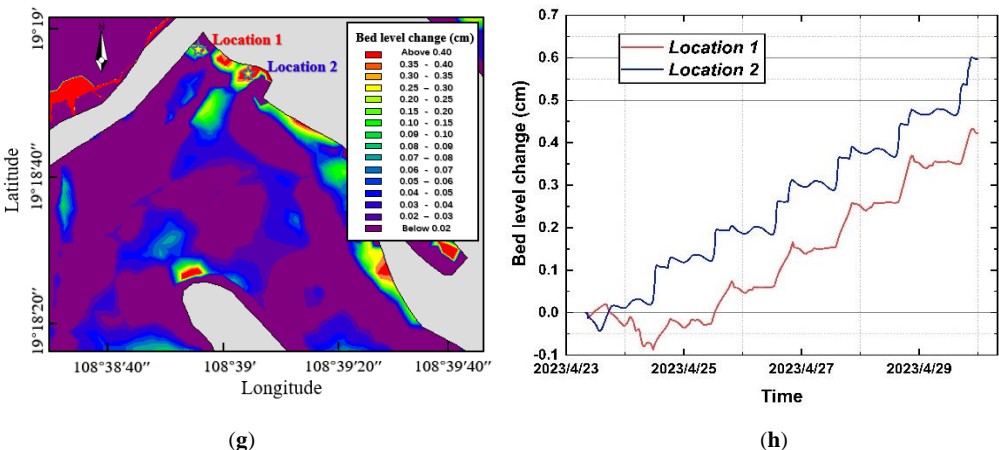

(g)                                      (h)

Figure 13 Bed level change of deposition in Danchangcun

From April 27th to 30th, an overall increase in deposition thickness was noted, reaching approximately 0.59 cm. Two rapid deposition phases were identified: the first, on April 23rd from 13:30 to 20:30, coincided with astronomical mid-tide but exhibited lower current velocities than expected, as per ADCP 01 measurements. The second phase followed an spring tide on April 22nd, which stirred turbulent currents and enhanced scouring, leading to increased sediment concentration in the estuary. The tide on April 23rd was moderate, significantly reducing current velocity and sediment transport capacity, resulting in sediment deposition in the estuary.

On April 27th, during astronomical neap tide, lower water levels and reduced tidal ranges led to slower currents, enhancing sedimentation and weakening lateral erosion. The current's reduced capacity limited the transport of larger sediment particles, allowing only fine grains to settle at the water's bottom. These findings underscore the complex interplay between sediment deposition and erosion in estuarine environments and highlight the influence of tidal dynamics on sediment transport processes.



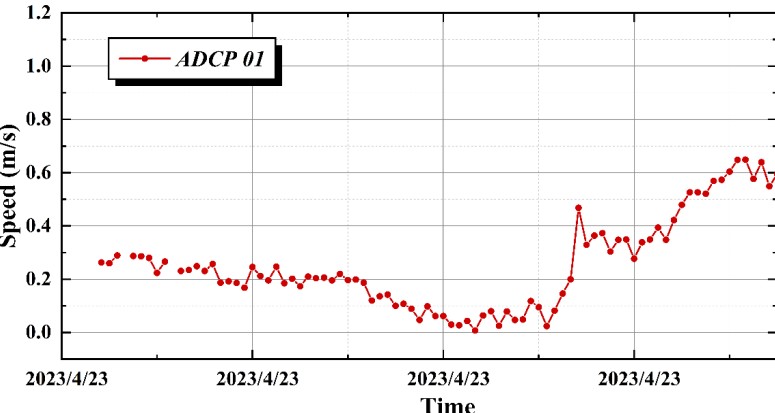


Figure 14 Current speed on April 23rd

In the Xiantiancun estuary, sediment deposition is influenced by its narrower configuration
compared to Danchangcun, with numerous tributaries contributing to a dispersed flow and
reduced kinetic energy. This results in variable sediment deposition levels at the entrances of the
tributaries, although the overall deposition is less extensive than at the Danchangcun mouth. The
maximum observed deposition thickness within the estuary is 0.58 cm at Location 2, while other
areas exhibit thicknesses between 0.3 cm and 0.5 cm.
Two significant deposition sites are located near the sand mouth, which may facilitate the
mouth's further expansion. Additionally, a substantial, albeit thin, silting zone is identified at the
rear of the river island (Location 1), covering a considerable area. These findings indicate the
complex interplay of sedimentary processes in estuarine environments and the potential for
morphological changes due to deposition patterns.

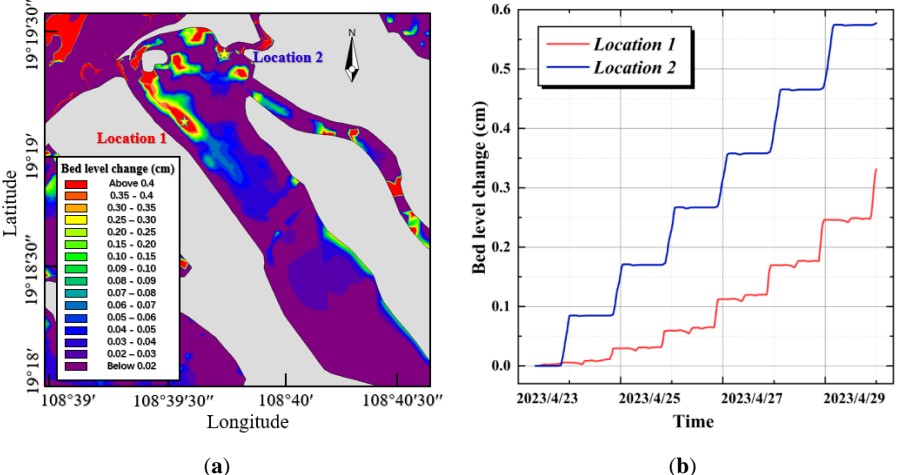

Figure 15 Deposition in Xiantiancun: (a) shows changes of sedimentation thickness of Xiantiancun with
palette; (b) shows changes of sedimentation thickness of Xiantiancun in detail.

To summarise, the Changhua River estuary exhibits distinct sedimentation patterns, with
notable deposition occurring in both the estuary and slender channel regions. The estuary
depositions are a result of interplay between hydrodynamic conditions and geological settings.
Specifically, the estuary is subject to persistent northeast-southwest coastal currents and wave
action, leading to the formation of a two-way sand mouth that constricts the estuary's width. The
sandy, unstable riverbed further contributes to substantial sediment deposition due to the reduced
gradient and sediment transport capacity of the fluctuating discharge. This has, over time, led to
the formation of river islands, transforming the estuary into a complex channel system with
multiple small estuaries. The main channel, situated between these islands, experiences shallow
water depths during low tide.
In the Danchangcun region, the estuary displays a maximum sediment deposition thickness
of 0.59 cm. The presence of a small river island in this area results in shallow deposition near the
island, with some areas having thicknesses below 0.3 cm. In contrast, deeper deposition is
observed along the riverbanks and particularly near the estuary. The sand mouth at the estuary's





entrance is influenced by river erosion and coastal currents, leading to the formation of a new
small sand mouth to the southwest. The original sand mouth tends to thicken after fracturing,
with scouring at its front end and deposition at the fractured end, reaching a maximum thickness
of 0.6 cm. This suggests that the estuary's current is obstructed by multiple depositional strips,
resulting in a slower current and increased deposition.
In the Xiantiancun region, the estuary is narrower than in Danchangcun, with numerous
tributaries dispersing the flow and reducing energy. This leads to varying degrees of deposition at
the entrances of the tributaries, although the overall deposition is less than that observed at the
Danchangcun mouth. The maximum deposition thickness at the estuary reaches 0.58 cm, with
other areas exhibiting thicknesses ranging from 0.3 cm to 0.5 cm. Deposition near the sand
mouth contributes to its expansion, and a long silting zone is present at the rear of the river island,
characterized by a thin layer over a large area.

## 3.6 Analysis of deposition in Changhua River channel

Changhua River's channel exhibits two key sediment deposition sites: the Chahe confluence
and an area near Jiuxiancun. These areas are prone to significant sedimentation as the river
narrows from a wide estuary to a more confined channel, increasing the risk of blockages(Figure
16a). The primary sedimentation zone is located on the right bank of the distributary, with the
maximum thickness measuring 0.47 cm (Figure 16b). Deposition is most intense around the river
island and decreases from the right side towards the rear and the left side of the island. This
distribution suggests that sedimentation is more pronounced in the upper, narrower section of the
channel.
In the main channel, erosion occurs on the ocean-facing right side, while the left side is



subject to deposition. The sediments on the left bank are likely sourced from tidal actions or
upstream inflows, a process that requires further study. The lateral variation in sedimentation and
scouring highlights the intricate sediment dynamics within the river channel.

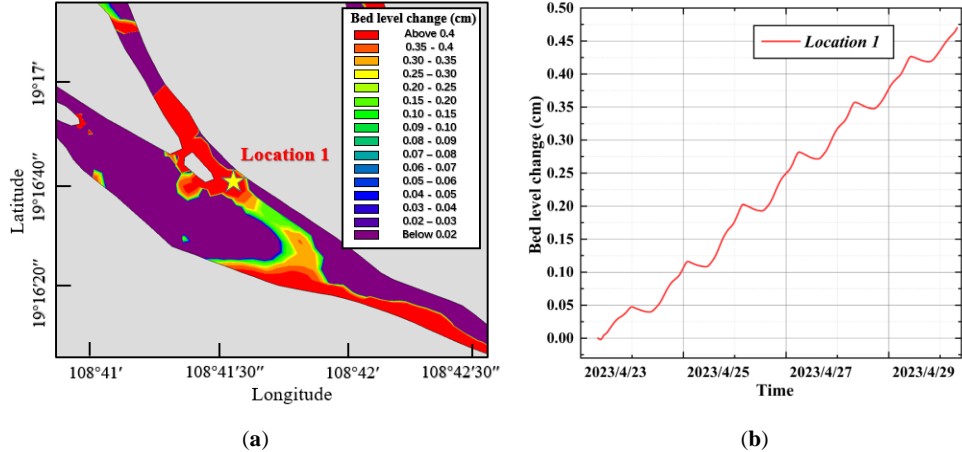

(a)                                         (b)

Figure 16 Deposition in channel: (a) shows changes of sedimentation thickness of channel with palette; (b)
shows changes of sedimentation thickness of channel in detail.
Analysis of topography and flow velocities along the river island banks indicates a pattern
of alternating unidirectional and counter-currents (Figure 17). The current speeds peak at 0.21
m/s during opposing flows and reach approximately 0.68 m/s when currents are in the same
direction. The Xiantiancun section, marked by a constricted channel and intensified currents, is
prone to sediment accumulation. As tides recede, the river's hydrodynamic energy weakens,
facilitating the convergence of the Xiantiancun course with the estuary's incoming flows. This
interaction leads to the predominant deposition of sediment on the left bank of the main channel,
facing the ocean, which is influenced by high-tide influxes.





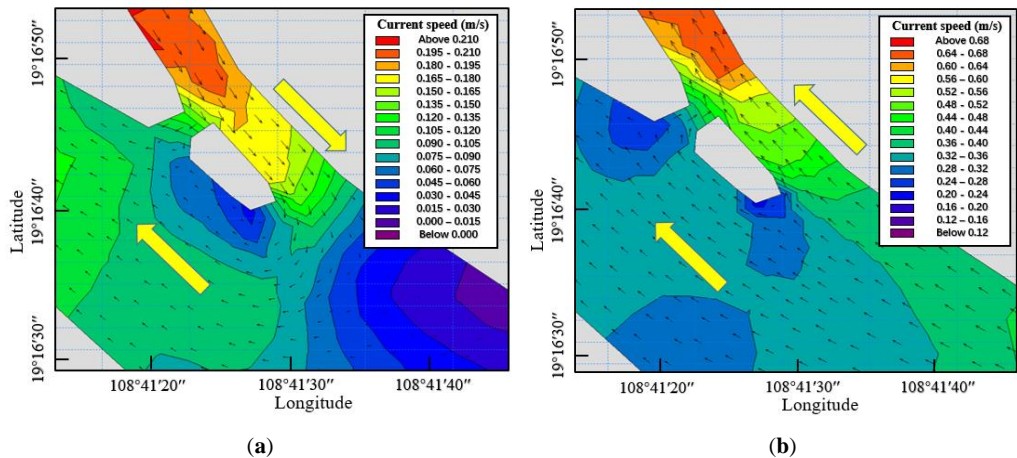

(**a**)                                        (**b**)

Figure 17 Flow around the river island: (a) shows the flow around river island in opposite directions; (b) shows the flow around river island in same directions

A secondary sediment deposition site has been identified in proximity to Jiuxiancun, with the maximal sediment thickness measuring 0.81 cm (Figure 18). This deposition zone is elongated and in close proximity to the coast, while erosion is observed on the opposing bank. The river's erosive action has led to the removal of the opposite bank, with the displaced sediment accumulating near Jiuxiancun. Over time, this accumulation is expected to enhance the river bend's curvature, potentially hindering the river's natural evolution.

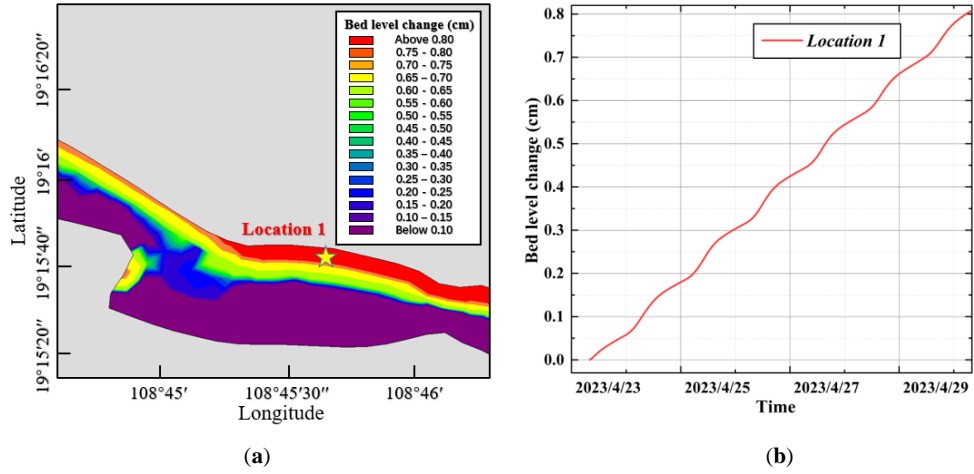

(**a**)                                        (**b**)

Figure 18 Deposition near the Jiuxiancun: (a) shows changes of sedimentation thickness of Jiuxiancun with palette; (b) shows changes of sedimentation thickness of Jiuxiancun in detail.



To summarise, there are two clear deposits at the channel of the Changhua River. One
occurs at the intersection of Xiantiancun and Danchangcun, while the other is near Jiuxiancun.
Compared to the fork, sedimentation near the Jiuxiancun is deeper and thicker. The final
deposition thickness of the model is 0.81 cm. The fork was deposited near the river island, and
the simulation resulted in a displacement of 0.47 cm. The sediment carried by the high tide may
be the source.

## 4. Measurements for sediment regulation

### 4.1 Anticipated Regulation Measures for Estuary Siltation

According to the previous analysis, the Changhua River estuary is controlled by tides, and
there is a long-term repeated coastal flow. Therefore, the current drives the sediment in the river
bed to form a composite channel. According to the results of the sediment transport model, the
main sediment deposition near the estuary occurs in Danchangyuan Village and Xiantian Village.
Based on the formation mechanism of estuarine sediment, the following two measures are put
forward.
The slope protection of dikes within tidal estuaries necessitates an engineering approach
that prioritizes resilience against environmental impacts, structural integrity, and effective wave
dissipation. Additionally, these measures should exhibit longevity and ease of construction,
maintenance, and management. For dikes with extensive beachfront areas, the strategic planting
of wave-resistant vegetation such as forests or reeds can significantly mitigate wave impact.
Furthermore, in the intertidal zone, the cultivation of mangroves or Spartina species, tailored to
local climatic and salinity conditions, can foster a vegetative buffer that aids in wave reduction



and promotes sediment deposition.
In the context of the Changhua River Estuary, the persistent northeast to southwest coastal
currents have resulted in a bi-directional sandbar formation, exacerbating the estuary's
constriction. Given the sandy nature of the estuary's bed and the associated poor riverbed
stability, traditional port protection engineering is challenging to implement. Consequently,
regular dredging of the sandbar at the estuary's mouth is imperative, complemented by stringent
control of upstream inflow. It is also suggested that a designated sedimentation area and multiple
contingency flow paths be established within the estuary to accommodate long-term river
discharge into the sea.

## 4.2 Anticipated Regulation Measures for River Channel Siltation

An objective analysis of sediment content in the river channel is needed to further assess the
situation. The tables below provide statistics for the annual average sediment transport and
sediment concentration at the Baoqiao station in the lower reaches of Changhua River.
**Table 6** **Statistical Table of Annual and Monthly Sediment Transport Rate at Baoqiao Station (kg/s)**

| year | January to April | May | June | July | August | September | October | November | December |
|------|------------------|-----|------|------|--------|-----------|---------|----------|----------|
| 2013 | 0 | 5.56 | 7.03 | 9.69 | 120 | 14.3 | 9.42 | 11.4 | 0 |
| 2014 | 0 | 11.0 | 7.80 | 217 | 53.0 | 108 | 23.4 | 5.68 | 0 |
| 2015 | 0 | 7.91 | 6.62 | 3.95 | 5.02 | 8.91 | 18.4 | 1.47 | 0 |
| 2016 | 0 | 0.397 | 0.737 | 1.30 | 259 | 80.3 | 90.4 | 14.6 | 0 |
| 2017 | 0 | 11.4 | 19.0 | 12.9 | 10.5 | 5.69 | 30.5 | 6.99 | 0 |
| 2018 | 0 | 8.00 | 18.7 | 83.4 | 373 | 95.9 | 11.0 | 7.05 | 0 |
| 2019 | 0 | 4.76 | 0.370 | 0.430 | 6.68 | 6.90 | 1.19 | 0.054 | 0 |
| 2020 | 0 | 0.100 | 3.35 | 0.066 | 0.429 | 0.117 | 15.7 | 3.73 | 0 |



| | | | | | | | | |
|---|---|---|---|---|---|---|---|---|
| 2021 | 0 | 0.799 | 0.851 | 0.417 | 0.831 | 6.65 | 91.7 | 0.339 | 0 |
| annual mean of 2013-2021 | 0.00 | 5.55 | 7.16 | 36.6 | 92.1 | 36.3 | 32.4 | 5.70 | 0.00 |

**Table 7 Statistical Table of Annual and Monthly Sediment Concentration at Baoqiao Station (kg/m³)**

| Year | January to April | May | June | July | August | September | October | November | December |
|---|---|---|---|---|---|---|---|---|---|
| 2013 | 0 | 0.047 | 0.080 | 0.092 | 0.318 | 0.060 | 0.141 | 0.081 | 0 |
| 2014 | 0 | 0.046 | 0.063 | 0.772 | 0.270 | 0.319 | 0.172 | 0.068 | 0 |
| 2015 | 0 | 0.071 | 0.065 | 0.056 | 0.060 | 0.127 | 0.182 | 0.055 | 0 |
| 2016 | 0 | 0.045 | 0.068 | 0.068 | 0.562 | 0.238 | 0.246 | 0.085 | 0 |
| 2017 | 0 | 0.081 | 0.101 | 0.108 | 0.086 | 0.084 | 0.258 | 0.107 | 0 |
| 2018 | 0 | 0.079 | 0.087 | 0.334 | 0.633 | 0.224 | 0.099 | 0.064 | 0 |
| 2019 | 0 | 0.026 | 0.007 | 0.008 | 0.049 | 0.041 | 0.026 | 0.005 | 0 |
| 2020 | 0 | 0.007 | 0.060 | 0.005 | 0.011 | 0.004 | 0.101 | 0.032 | 0 |
| 2021 | 0 | 0.008 | 0.014 | 0.017 | 0.017 | 0.064 | 0.417 | 0.007 | 0 |
| annual mean of 2013-2021 | 0.00 | 0.046 | 0.061 | 0.162 | 0.223 | 0.129 | 0.182 | 0.056 | 0.00 |

Hainan Island has a tropical monsoon climate with heavy rains in summer (Mao et al.,
2006). About 77% of the annual rainfall is concentrated in the wet season from May to October.
Analysis of the table reveals that sediment in the lower reaches of Changhua River primarily
originates during the summer storm surge period. The measurable sediment concentration is
concentrated from May to November, with a peak in July and August. The average sediment
transport and concentration peaked in August from 2013 to 2021. Following the government's
decision in 2018 to entrust Hainan River Channel Comprehensive Remediation Engineering Co.,
Ltd. with dredging work on the Changhua River channel, the monthly average sediment transport
and concentration at the Baoqiao station experienced a sharp decrease. The significant
improvement in sediment deposition is evident, with the average sediment transport in August
decreasing from 373 kg/s in 2018 to 0.831 kg/s in 2021. Similarly, the average sediment



concentration in August decreased from 0.633 kg/m$^3$ in 2018 to 0.017 kg/m$^3$ in 2021.
The remediation of substantial siltation can be categorized into protective engineering and
dredging strategies. For protective measures, reference should be made to the "Code for Design
of River Regulation". In the lower reaches of the Changhua River, a bifurcated section presents
unique challenges and opportunities. As per the principles outlined in the aforementioned code,
regulation measures should be employed to stabilize the bifurcated reach when it is in a
favorable developmental state for economic and social advancement. To this end, regulatory
structures can be strategically positioned at the bifurcation's upstream node, the river's inlet, local
scour sections within the river's bends, and at the extremities of Jiang Xinzhou. The specific type
of regulation project at the exit of the branch road should be selected based on the prevailing
conditions. For instance, beach preservation can be facilitated through afforestation, while bank
stabilization can be achieved through the implementation of protective works.
For the lower reaches of the Changhua River, three distinct schemes have been proposed:
Scheme 1: Given the predominance of sand and silt in the area, it is proposed to undertake
government-supervised artificial sand excavation to dredge the river. This approach must adhere
to several stipulations:
1.   Clearly define the annual management requirements for planned exploitable areas.
2.   Investigate the current state of sand mining management, identify key issues, and propose
the establishment of sand mining management institutions, along with measures for
improvement and funding requirements.
3.   Propose dynamic monitoring and management strategies for exploitable areas and river
sections affected by sand mining, tailored to the characteristics of each river.
Scheme 2: Considering the Changhua River's unique strip-shaped sedimentary landform



with alternating sandbar and lagoon deposits, a divide-and-conquer approach is suggested.
Historically, efforts to control coarse sediment have been characterized by a "blocking" strategy,
utilizing soil and water conservation methods and reservoirs to prevent coarse sediment from
entering the river. However, empirical evidence suggests that coarse sediment inevitably moves,
necessitating a more effective strategy. The proposed divide and conquer method involves
separating coarse sediment from fine sediment, transporting the medium and fine sediment
through the river channel, and managing the coarse sediment with the aid of desilting
engineering facilities, such as innovative self-desilting corridors.
Scheme 3: Recognizing the distinct flood (May-October) and dry (November-April) seasons
of the Changhua River, with the flood season accounting for 77% of the annual flow, it is
proposed to establish seasonal gates within the river. These gates can control the flow by
adjusting the number and operation mode of inlets and outlets. Additionally, grab dredgers can be
utilized to assist in river dredging during the flood season.
## 5. Conclusions
The study successfully applied a wave-current coupled sediment transport model to the
lower reaches of the Changhua River in Hainan Island. By integrating field measurements,
remote sensing techniques, and the Van Rijn model, this research has developed a comprehensive
model capable of accurately simulating sediment behavior under the combined action of waves
and currents. The following conclusions reflect a robust understanding of the study's themes:
Model Validation and Effectiveness: The sediment transport model has been rigorously
validated using both theoretical and empirical methods. The theoretical validation was conducted



using the sediment transport rate method, while empirical validation involved comparing the
model's simulated suspended sediment concentration (SSC) with observed data from the Baoqiao
Station. The model demonstrated a high degree of accuracy with an NSE value of 0.8389,
indicating excellent agreement between observed and simulated SSC values.
Deposition Patterns: The study reveals the deposition patterns in the estuary and
downstream river channel of the Changhua River, which are closely related to the interplay
between hydrodynamic conditions and geological settings. Specifically, the estuary's deposition
is primarily influenced by the northeast-southwest coastal currents and wave action, while the
river channel's deposition is associated with the river's constriction and changes in flow velocity.
Spatial Variability of Sediment Properties: The study underscores the importance of
considering the spatial variability of sediment properties. Sediment parameters obtained through
direct sampling are crucial for enhancing the model's accuracy, which is more effective than
relying on empirical formulas or theoretical calculations.
Limitations of Model Application: Despite the successful operation of the model in this
study's case, there are limitations. Many models rely on empirical formulas derived from specific
experimental conditions or field observations, which may limit the model's applicability in
different environments or under varying wave and current conditions. Additionally, models may
not fully account for the impact of human activities (such as dredging, coastal engineering, river
diversion, etc.) on sediment transport.
In summary, this study not only validates the effectiveness of the wave-current coupled
sediment transport model in the downstream reaches of the Changhua River but also provides
robust scientific evidence for sediment management and coastal evolution in similar downstream
river environments. Future research should further consider the impact of human activities and



explore the applicability of the model under different environmental conditions to enhance its
accuracy and expand its range of application.

## 674    Data Availability

This study utilized shoreline data obtained free from the Geophysical Data System (GEODAS) at
https://www.ngdc.noaa.gov/mgg/gdas/gx_announce.Html; The wind field data are available from
European    Centre    for    Medium-Range    Weather    Forecasts    (ECMWF)    at
https://cds.climate.copernicus.eu/cdsapp#!/dataset/reanalysis-era5-single-levels?tab=form; In this
study part topographic data was obtained from the ETOPO1 dataset, developed by NOAA, which
includes comprehensive bathymetric and topographic information. The dataset has a resolution of
1    arc-minute    and    is    widely    used    for    various    geophysical    applications."    [DOI:
10.7289/V5C8276M]; Topographic data measured by ADCP and hydrological station data that
support the findings of this study are available from Haikou Marine Geological Survey Center but
restrictions apply to the availability of these data, which were used under license for the current
study, and so are not publicly available. Data are however available from the authors upon
reasonable request and with permission of Haikou Marine Geological Survey Center.

## 687    Author contribution

**Yuxi Wu:** Writing – review & editing, Writing – original draft, Visualization, Validation, Software,
Resources, Methodology, Investigation, Formal analysis, Data curation, Conceptualization. **Enjin**
**Zhao:** Writing – review & editing, Writing – original draft, Supervision, Resources, Project
administration, Conceptualization. **Xiwen Li:** Investigation (data collection), Validation,
Supervision, Project administration. **Shiyou Zhang:** Investigation (data collection), Validation.

## 693    Competing interests



The contact author has declared that none of the authors has any competing interests.

## Acknowledgments

Acknowledgment section of the article contain the following:
The authors are grateful to Haikou Marine Geological Survey Center, China Geological Survey
for providing data and technical support. Technical assistance during the field measurement
campaign is duly acknowledged.
The authors are grateful to Copernicus Climate Change Service: ERA5 hourly data on single
levels from 1940 to present (2021-2023) was downloaded from the Copernicus Climate Change
Service (2024).
The authors are grateful to Hainan Geographic Information Public Service Platform: The map in
this paper is quoted from Map World • Hainan (https://hainan.tianditu.gov.cn/), a website
developed by Hainan Geographic Information Public Service Platform.

## Funding

The text ends with an acknowledgment section and statement that includes:
● National Natural Science Foundation of China (Grant Nos. 52371295,52001286),
● Guangdong Basic and Applied Basic Research Foundation (Grant Nos. 2022A1515240002),
● Hubei Provincial Natural Science Foundation of China (Grant No. 2023AFB576)



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
