# Peer review of "Application of Wave-current coupled Sediment Transport Models with Variable Grain Properties for Coastal Morphodynamics: A Case Study of the Changhua River, Hainan"

_EGUsphere, 2024_

## Referee Comment (RC1)

**Review of the manuscript**

**'Application of Wave-Current coupled Sediment Transport Models with Variable Grain properties for Coastal Morphodynamics: A Case study of the Changhua River, Hainan'**

**by Wu et al.2024**

In this paper, the authors investigate erosion and deposition in the lower reaches of the Changhua River Estuary by implementing hydrodynamic and sediment transport modelling. They attempt to highlight the waves' influence on sediment dynamics in rivers with low sediment concentration. They make use of a wave model that incorporates the Van Rijn formula for the calculation of bed load and suspended sediment transport. They also consider various sediment properties both at the estuary and the main river channel. After model calibration, they examine erosion and sedimentation at several locations.

The manuscript is not very well written and critical information is missing especially regarding their model's set up. Because of this deficiency, it is very difficult to judge the validity and robustness of the modelling, and the results and conclusions cannot be really assessed unless the authors provide more information and improve the quality of their manuscript including their figures. There are also some fundamental issues that I am going to list here:

**Main comments on the hydrodynamic model**

1. The authors claim to use a three-dimensional model (lines 131-132) while the equations they present are depth-averaged two dimensional in x and y. It is also never mentioned which model the authors use. Is it a well established and validated one? Is it open-source like e.g., Delft3D and TELEMAC? is it an in-house one? Please provide this information and where and if it has been implemented for similar applications.

2. Not enough information is given regarding the wave model parameters. Please give details on how the wave radiation stresses are calculated in the model and how the waves are incorporated in the model equations i.e., via the mentioned JONSWAP spectrum, the frequency of occurrence of different wave heights and directions and number of days in different seasons and the wave fixed parameters at the open boundary (lines 295-300).

3. Although the simulation period is never explicitly mentioned in the manuscript, it seems that this coincides with the calibration period (23/04-30/04/2023). To be validated though, the model needs to run with the same setup for another period. In addition, there is no mention in the manuscript of the calibration parameters (e.g., roughness, diffusion, viscosity etc.) and how these were modified. For example, do the authors use uniform values or spatially varying ones?

4. The model grid is not presented, is it structured or unstructured? Is it regular or curvilinear? What is the model's resolution? The limits of the white rectangular in Figure 6 do not coincide with the grid coordinates as given in Line 280. The authors claim in lines 286-289 that the resolution of the ETOPO1 data is not sufficient for this research however it seems that these are the data used for developing the bathymetry of their model as it is written in the legend of Figure 6. The time step is also missing.

5. The hydrodynamic results (section 3.3) are ill-presented and not adequately analysed. At which moment the results given in Figure 9 correspond to? Are these depth and/or time

averaged? The depicted current of Figure 9 is in some instances referred to as a tidal (line 345) and in others as a coastal one (line 430). Are these averaged, combined tide and wave currents over this week of simulation or are taken at a certain time of calculation and if so, which one? Furthermore, it is written that the results are given at the times of high and low water but tidal currents at these times are minimal. What's the situation at slack times? Finally, there is no mention in the manuscript of whether we are looking at a period of neap or spring tide. Water levels in Figure 7 remain almost constant during the simulation period and in any case a spring-neap cycle takes place within 15 days and not one week (lines 473-478).

6. The authors underline the fact that the tide is the most prominent effect at the point of comparison against wind and waves (lines 335-336) but under these circumstances, their validation process cannot be considered as a validation of the combined effect of current-wave but only of tide. Therefore, a case where all factors are important is required to demonstrate the relative capability of the model.

**Main comments on the sediment transport model**

1. The manuscript misses information on the initial and boundary conditions and rates for both bed load and suspended sediments transport. The sediment motion equation implemented in the model (line 363) needs to be presented. How are the two modules coupled? Is it an online or offline coupling?

2. As mentioned earlier, the model's given equations (1)-(4) in the paper are two-dimensional depth averaged. How is then the velocity vertical profile computed in equation (7)?

3. There is a serious confusion in section 3.4.1. The authors calculate a uniform over space sediment thickness from their model equal to 4.1 cm. If what they write in lines 365-366 is right, then they get results after the one-week simulation that they did for 23/04/2023-30/04/2023. But then they use the estuary's sediment discharge of 2022 (line 389-390) and they compare their model result with field data covering one month (July 2022)!! Not surprisingly then, there is a serious discrepancy of 1.1 cm between the model and the observations which is too big to be ignored. Besides, the assumption that the uniform over space result from a one-week simulation for a certain year can be representative of any year for that specific area is too crude.

4. From what is written in section 3.4.2, it seems that the authors not only have sediment data for an entire month (July 2022) but that they run their model for this period as well. Why don't they do then the comparison based on results from this simulation? In any case, the fact that they present results analysis for different time periods is already problematic.

**General comments**

1. In their figures, the authors fail to provide all the necessary information regarding their study case. They refer to stations that can be nowhere found in their figures and so the reader cannot understand from which location the results and data are taken? Specifically, Dongfang Ocean station, Danchangcun, Xiantiancun and Jiuxiancun stations cannot be found in the figures. The same for the ADCP.

2. Chapter 4 does not discuss the scientific output and fails to explore and highlight the importance of the outcomes. It only provides a list of potential measures to be taken against siltation which have only a local interest. These can be hardly supported

considering that they are based on conclusions from not adequately justified modelling. I suggest that the authors rewrite this section after they have addressed all the raised issues in their modelling approach but from a non-local perspective.

3. The conclusions read more like a summary. What are the key findings of this research, which messages can we take from this and why these would contribute further to the research on this topic?

---

## Author Comment (AC1)

Dear reviewer,

On behalf of my co-authors, we thank you for giving us a chance to revise and improve the quality of our article.

We have read your comments carefully and have made revision. We have tried our best to revise our manuscript according to the comments: "Application of Wave-current coupled Sediment Transport Models with Variable Grain Properties for Coastal Morphodynamics: A Case Study of the Changhua River, Hainan (egusphere-2024-2154)".

The main revisions in the new manuscript are:

1. The parameters of the hydrodynamic model have been presented in a table.

2. Initial conditions for the sediment model and the wave model's open boundaries have been included.

3. Section 3.4.1 has been removed.

4. Additional details regarding the hydrodynamic simulation results have been provided in Section 3.3.

5. The locations of the relevant stations and geographical positions required within the text are now illustrated with figures.

6. Chapter 4 has been thoroughly rewritten.

7. Conclusions has been revised.

Here is a point-by-point response to the comments and concerns.

Thank you for taking the time to consider our research and we look forward to hearing from you at your earliest convenience.

Sincerely,

Yuxi Wu

China University of Geosciences, Wuhan,

Wuhan 430074, P.R.China

E-mail: yuxiwu@cug.edu.cn

**Detailed comments part:**

**Point 1: The authors claim to use a three-dimensional model (lines 131-132) while the equations they present are depth-averaged two dimensional in x and y. It is also never mentioned which model the authors use. Is it a well established and validated one? Is it open-source like e.g., Delft3Dand TELEMAC? is it an in-house one? Please provide this information and where and if it has been implemented for similar applications.**

Response: Thank you for pointing out the discrepancy in our manuscript regarding the model description. We sincerely apologize for the error in the equations presented and appreciate the opportunity to clarify and correct this. Equations (1) to (4) have been removed.

We have made the necessary corrections in the manuscript to accurately reflect that we are using the FVCOM (Finite Volume Coastal Ocean Model), a well-established and widely used three-dimensional model for simulating coastal and oceanographic processes. FVCOM source code was obtained from the Marine Ecosystem Dynamics Modeling Laboratory (http://fvcom.smast.umassd.edu/). The FVCOM model is indeed open-source and has been extensively validated across a variety of marine and estuarine applications.

The revised sentence now reads as follows:

"…, *An unstructured grid, finite volume, regional ocean model FVCOM (Chen et al., 2003) was used to simulate the hydrodynamic background and hydrological features. It has been widely used for the study of coastal oceanic and estuarine circulation (Jiang and Xia, 2016; Huang et al., 2008; Lai et al., 2018; Chen et al., 2008)."*

Applications:

Jiang, L., Xia, M.: Dynamics of the Chesapeake Bay outflow plume: Realistic plume simulation and its seasonal and interannual variability. *Journal of Geophysical Research: Oceans*, 121(2): 1424-1445. doi: 10.1002/2015JC011191, 2016.

Chen, C., Liu, H., Beardsley, R. C.: An unstructured grid, finite-volume, three-dimensional, primitive equations ocean model: application to coastal ocean and estuaries. *Journal of atmospheric and*

*oceanic    technology*,    20(1):    159-186.    doi:    10.1175/1520-0426(2003)020<0159:AUGFVT>2.0.CO;2, 2003.

Chen, C., Xue, P., Ding, P., Beardsley, R. C., Xu, Q., Mao, X., Gao, G., Qi, J., Li, C., Lin, H., Cowles, G., Shi, M.: Physical mechanisms for the offshore detachment of the Changjiang Diluted Water in the East China Sea. Journal of Geophysical Research: Oceans, 113(C2). doi: 10.1029/2006JC003994, 2008.

Huang, H., Chen, C., Blanton, J. O., Andrade, F. A.: A numerical study of tidal asymmetry in Okatee Creek, South Carolina. Estuarine, *Coastal and Shelf Science*, 78(1): 190-202. doi: 10.1016/j.ecss.2007.11.027, 2008.

Lai, W., Pan, J., Devlin, A. T.: Impact of tides and winds on estuarine circulation in the Pearl River Estuary. *Continental Shelf Research*, 168, 68-82. doi: 10.1016/j.csr.2018.09.004, 2018.

**Point 2: Not enough information is given regarding the wave model parameters. Please give details on how the wave radiation stresses are calculated in the model and how the waves are incorporated in the model equations i.e., via the mentioned JONSWAP spectrum, the frequency of occurrence of different wave heights and directions and number of days in different seasons and the wave fixed parameters at the open boundary (lines 295-300).**

Response: We have now included detailed information about the wave model parameters and how wave radiation stresses are calculated. The wave model employed in this study is the open-source Simulating WAves Nearshore (SWAN) model, which is widely recognized for its accuracy in wave action calculations.

The wave radiation stresses in the SWAN model are computed by solving the wave action conservation equation, taking into account wave nonlinearity and spectral peak shifting. At the

open boundaries, we have set up the model using the JONSWAP spectrum, which is a well-established spectrum for representing the statistical properties of wind-generated waves in deep water. The specific parameters for the JONSWAP spectrum have been added in the appropriate section of the manuscript. To ensure the accuracy of the wave boundary conditions, we have adjusted the parameters based on multi-year wave data from the Dongfang Station. The directional resolution is set to 40 sectors, with a particular focus on the southwest (SSW) and southwest (SW) directions where the waves are most frequent.

The revised sentence now reads as follows:

"*Considering the limitations of the FVCOM model in wave calculations, this study selects the widely-used third-generation SWAN model (SWAN team, 2006)for numerical simulation of wind waves in this region. The wave field are driven by wind and current from hydrodynamic model. The parameters used in the model setups are based on the values listed in Table 6. The wave model at the open boundary is defined by the JONSWAP spectrum, with a spectral resolution of 40 frequency bins and 36 directional sectors. Calibrate the parameters using multi-year wave data from the Dongfang Ocean Station. The directional resolution is set to 40 sectors, with a particular focus on the southwest (SSW) and southwest (SW) directions where the waves are most frequent. The wind speed and wind direction are from the ERA5 reanalysis data provided by ECMWF. The peak parameter ($\gamma$) of the JONSWAP spectrum, indicative of the wave asymmetry, was specified at 3.3, and the spectral width parameter was set to 0.07 to define the shape of the wave spectrum.*

Table 6 Parameters of the wave model

| Parameter | Value |
| --- | --- |
| Whitecapping dissipation ($C_{ds}$) | $2.36 \times 10^{-5}$ |
| Pierson–Moskowitz ($S_{pm}$) | $3.02 \times 10^{-5}$ |
| Dissipation (alpha) | 1.0 |
| Breaking index (gamma) | 0.73 |
| JONSWAP formulation ($C_{bottom}$) | 0.067 |

"

**Point 3: Although the simulation period is never explicitly mentioned in the manuscript, it seems that this coincides with the calibration period (23/04-30/04/2023). To be validated though, the model needs to run with the same setup for another period. In addition, there is no mention in the manuscript of the calibration parameters (e.g., roughness, diffusion, viscosity etc.) and how these were modified. For example, do the authors use uniform values or spatially varying ones?**

Response: Thank you for your insightful comment regarding the calibration parameters and the simulation period of our model. In response to your feedback, we have now included a comprehensive description of the calibration parameters in our manuscript. The calibration is performed using water level data from the Basuo Port from April and May of 2022. We have added a new table in the "Model Region and Settings" chapter, where we detail the calibration parameters such as roughness, diffusion, and viscosity. These parameters were adjusted using the control variable method, which involves a systematic approach to modify one parameter at a time while keeping others constant to observe and quantify its impact on the model output.

The revised sentence now reads as follows:

*"In this study, we calibrated the hydrodynamic model using water level data from April and May of 2022. The data were collected from Basuo port station. One-At-a-Time (OAT/OFAT) method (Czitrom, 1999) is used to modify the parameters, an effective local sensitivity analysis technique. In each experiment, we alter one factors while holding the others constant. During the calibration process, our primary focus was on the model's hydrodynamic response. This was achieved by adjusting the flow resistance parameters and the bed roughness coefficients within the model. The calibrated model parameters are presented in Table 5.*

Table 5 Parameters of the hydrodynamic model

| Parameter | Value |
|---|---|
| Shoreline | GSHHS |

| | |
|---|---|
| Bathymetry | ETOPO1 and ADCP in-situ |
| Grid | 0.25 km at the boundaries to 25 m near the coastline |
| Time period | 23/4/2023 00:00-30/4/2023 00:00 (Spring neap tide)
28/6/2022 00:00-1/8/2022 00:00 (High water period) |
| Manning number | 28 |
| Eddy viscosity | Smagorinsky formulation data 0.28 m²/s |
| Time step | 300 s |
| Tidal constituents | M2, S2, K1, O1, N2, K2, P1, Q1 |
| Wind/Sea level Pressure | ERA 5 |
| Validation | Basuo Port Station (19°06' N, 108°37' E)
ADCP 01, ADCP 02 |

"

**Point 4: The model grid is not presented, is it structured or unstructured? Is it regular or curvilinear? What is the model's resolution? The limits of the white rectangular in Figure 6 do not coincide with the grid coordinates as given in Line 280. The authors claim in lines 286-289 that the resolution of the ETOPO1 data is not sufficient for this research however it seems that these are the data used for developing the bathymetry of their model as it is written in the legend of Figure 6. The time step is also missing.**

Response: We sincerely apologize for the oversight regarding the model grid details and the associated errors in Figure 6. Your feedback is invaluable, and we have taken immediate steps to rectify these issues and enhance the clarity of our manuscript.

**Model Grid Details**: We have now included a detailed description of the model grid in the "Model Regionand Configuration" section of the manuscript. The Finite Volume Coastal Ocean Model (FVCOM) utilizes a non-structured triangular grid, which allows for flexible resolution and better adaptation to complex coastal geometries. The specific details of the grid, including the number of nodes and elements, are as follows:

● Grid Type: Unstructured triangular grid

● Number of Nodes: 13,814

- Resolution: In the offshore region, the grid density is lower, with a resolution of 0.25 km, while the nearshore part of the open boundary has higher grid resolution. In the main research area near the river channel, the grid resolution is highest, reaching 25 m.

  The model's resolution varies from 14 minutes to 7 minutes, with higher resolution in the area of interest to capture the detailed coastal dynamics. The grid is refined near the coast and in the estuary to better resolve the complex flow patterns and sediment transport processes.

**Correction of Figure 6**: We have revised Figure 6 to accurately represent the model grid and its boundaries. The white rectangular limits now correctly coincide with the grid coordinates, providing a clear visual representation of the model domain and its spatial extent.

**ETOPO1 Data Clarification:** We acknowledge the confusion caused by the previous description of the ETOPO1 data usage. The ETOPO1 data, with a resolution of 1 arc-minute, was indeed used as a base to develop the bathymetry of our model. However, the scope of the model is small. The model's span is 14 arc-minutes by 7 arc-minutes. Therefore, we have conducted topographic surveys using an ADCP-equipped survey vessel. The measured data have been interpolated to generate the high-resolution terrain information used in our model. The specific locations of these survey points are now clearly indicated in Figure 6.

**Model Time Step:** The model was run with a time step of 300 seconds, which is suitable for capturing the rapid changes in water levels and currents in the coastal area.

The revised sentence now reads as follows:

 "*..., The model employs a triangular unstructured grid. To enhance computational accuracy and reduce computation time, the density of boundary nodes gradually decreases from nearshore to offshore. In the offshore region, the grid density is lower, with a resolution of 0.25 km, while the nearshore part of the open boundary has higher grid resolution. In the main research area near the river channel, the grid resolution is highest, reaching 25 m. The entire study area grid comprises a total of 13, 814 computational nodes…..*"

The revised figure now shows as follows:

[Figure]

Figure 6 (a) Scope of study area (the white frame) and wave observation (the red star) from Dongfang; (b) ADCP collection points on site; (c) Grids and boundaries (map origination: https://hainan.tianditu.gov.cn/)

**Point 5: The hydrodynamic results (section 3.3) are ill-presented and not adequately analysed. At which moment the results given in Figure 9 correspond to? Are these depth and/or time averaged? The depicted current of Figure 9 is in some instances referred to as a tidal (line 345) and in others as a coastal one (line 430). Are these averaged, combined tide and wave currents over this week of simulation or are taken at a certain time of calculation and if so, which one? Furthermore, it is written that the results are given at the times of high and low water but tidal currents at these times are minimal. What's the situation at slack times? Finally, there is no mention in the manuscript of whether we are looking at a period of neap or spring tide. Water levels in Figure 7 remain almost constant during the simulation period and in any case a spring-neap cycle takes place within 15 days and not one week (lines 473-478).**

Response: We have taken your comments seriously and have made substantial revisions to address the concerns raised regarding the hydrodynamic results presentation and analysis.

**Clarification on the Timing of Results:** We acknowledge the lack of clarity regarding the

timing of the results presented in Figure 9. We have now specified that Figure 9 corresponds to the peak flow moments during both ebb and flood tide cycles, which occur at the transition phases between high and low tide. The revised part now reads as follows:

"..., *Figure 10b and 10c depict the flow field outside the estuary of the Changhua River. Figure 10b shows the flow field at 23:00 on April 23, 2023, corresponding to the peak of the flood tide (Rapid flooding tide). At this time, the tidal current flows in a northeast direction with a maximum speed of 0.62 m/s. Figure 10c shows the flow field at 13:30 on April 24, 2023, corresponding to the peak of the ebb tide (Rapid ebb tide), where the tidal current flows in a southwest direction with a maximum speed of 0.75 m/s. Overall, the tidal currents outside the Changhua River estuary generally follow a northeast-southwest reciprocating pattern, with flood tides flowing northeast and ebb tides flowing southwest, parallel to the shoreline. The maximum ebb current is faster than the maximum flood current.*"

**Depth and Time Averaging:** The depicted currents in Figure 9 are indeed instantaneous but not time and depth averaged over the simulation period.

**Tidal Currents Distinction:** We have revised the manuscript to distinguish clearly between tidal and coastal currents. The revised sentence now reads as follows:

"..., *The predominant northeast-southwest **tidal** current direction and wave action, has led to the formation of a two-way sand mouth, further narrowing the estuary.....*"

**Simulation Period and Neap-Spring Tide Cycle:** The study area experienced an incomplete transition from a new moon to the first quarter moon between April 22, 2023, and April 27, 2023. During the new moon, a spring tide occurred, and during the last quarter moon, a neap tide occurred. This is not the 15-day spring-neap cycle in the strict sense. The spring-neap cycle refers to the period from one spring tide to the next spring tide, or from one neap tide to the next neap tide. Since April 22, 2023 (the third day of the lunar month) was during the new moon, it was the last day of the period of spring tide. April 27, 2023 (the eighth day of the lunar month) fell during the first quarter moon, hence the transition from spring tide to neap tide as described.

The added part (section 3.3) now reads as follows:

*"Figures 11a and 11b illustrate the flow field inside the estuary of the Changhua River. Figure 11a shows the flow field at 23:00 on April 23, 2023, corresponding to the peak of the flood tide. Inside Estuary A, due to the topography, a large counterclockwise circulation forms around the central island, accompanied by several smaller vortices, with the overall trend of tidal currents flowing southeast along the river channel. In Estuary B, ocean inflows meet with river flows from upstream, ultimately converging into Estuary C through the passage between B and C. In Estuary C, the flow is more unidirectional compared to A and B, with upstream water flowing into the ocean, then following the northeast-directed tidal current outside the Changhua River estuary. Figure 11b shows the flow field at 13:30 on April 24, 2023, at the peak of the ebb tide. At this time, the circulation inside Estuary A reverses to a clockwise direction, and other smaller vortices change direction accordingly, with the overall trend of tidal currents flowing from upstream to the ocean. In Estuary B, the dominant force is the high-speed flow from Estuary C, which enters B through the narrow passage between B and C, splitting into two opposite directions: one part flows into the ocean, and the other flows upstream, forming a circulation within the river channel. In Estuary C, the water flows upstream from the ocean along the river channel.*

[Figure]

Figure 11 Flow field inside the estuary :(a) moment of the maximum flood current; (b) moment of the maximum ebb current

*To further analyze the characteristics of the flow field in the study area, flow fields are selected for analysis during the transition from low tide to high tide and from high tide to low tide. Figure 12f depicts the location of the research area. Figure 12a shows the flow field at low tide, where the tidal current outside the estuary flows northeast, and water in the main river channel downstream of the Changhua River flows upstream from the ocean. After low tide (during flood tide), water flow velocity gradually increases, with the tidal current outside the estuary consistently flowing northeast. During this period, the main river channel maintains an eastward flow.*

*Figure 12b illustrates the moment of flow direction change during flood tide, when the flow direction outside the estuary rotates clockwise along the shoreline from the south (toward Beili Bay). The northern ocean current (outside Changhua Harbor) also begins to rotate clockwise, flowing into Estuary C, then into the ocean through the passage between B and C, forming a circulation that enhances the clockwise rotation of the northern ocean current. Subsequently, the flow direction gradually changes from northeast to southwest as it moves from the coast toward the open sea. The sand spit at the downstream estuary alters the flow direction and velocity. The sand spit can act as a natural barrier, causing the tidal current to change direction earlier during flood tide.*

*Figure 12c shows the flow field at high tide, where the tidal current outside the estuary has fully shifted to the southwest, while the flow direction further offshore is still transitioning. In the main river channel, the water flows from upstream toward the ocean. Estuaries B and C are influenced by the coastal current outside the northern part of the study area, flowing into the estuary opposite to Estuary A. After high tide (during ebb tide), the water flow velocity in the study area gradually increases, with the tidal current outside the estuary consistently flowing southwest. After some time, the water currents in the southern and northern parts of*

*the study area turn counterclockwise, and the flow direction in the B and C channels changes from inward to outward.*

*Figure 12d shows the flow field at the moment when the flow direction changes during ebb tide. It is evident that there are two counterclockwise circulations outside the Changhua River estuary: one from Beili Bay and the other from outside Changhua Harbor. The latter has a broader influence and thus plays a dominant role in determining the water flow direction in the study area, gradually shifting the coastal current from southwest to northeast. Figure 12e shows the flow field at low tide once again, where the water flow outside the estuary has shifted back to the northeast, repeating the previous flow pattern.*

*In summary, during the transition from flood to ebb tide, the flow field outside the estuary is driven by the deflection of water currents from Beili Bay and Changhua Port, shifting the flow direction from northeast to southwest. During the transition from ebb to flood tide, the deflection is primarily influenced by the circulation outside Changhua Port, shifting the flow direction from southwest to northeast. In channels A, B, and C within the study area, the flow direction changes are relatively consistent due to the passage between B and C. The flow direction in channel A aligns with the main river channel, flowing inward during flood tide and outward toward the ocean during ebb tide.*

[Figure]

Figure 12 Transition of the flow field (a-e) and location of the study area (f)

"

**Point 6: The authors underline the fact that the tide is the most prominent effect at the point of comparison against wind and waves (lines 335-336) but under these**

circumstances, their validation process cannot be considered as a validation of the combined effect of current-wave but only of tide. Therefore, a case where all factors are important is required to demonstrate the relative capability of the model.

Response: We sincerely apologize for the analysis error here. I have revised the description here, and the verification here is the verification of the hydrodynamic model.

The revised part now reads as follows:

*". The proximity of measurement point ADCP 01 to the land, coupled with its relatively shallow water depth, results in sea water being more susceptible to obstruction by the topography and friction from the seafloor at this location...."*

**Point 7: The manuscript misses information on the initial and boundary conditions and rates for both bed load and suspended sediments transport. The sediment motion equation implemented in the model (line 363) needs to be presented. How are the two modules coupled? Is it an online or offline coupling?**

Response: Thank you for your insightful comments and suggestions regarding the sediment transport model in our manuscript. We have taken your feedback seriously and have made the following revisions to address the concerns raised:

**Initial and Boundary Conditions:** We have now included detailed information on the initial and boundary conditions in Section 2.1 of the manuscript. The initial conditions for sediment concentration and the boundary conditions, including the influx of sediment at the upstream boundary, have been explicitly defined. The revised part now reads as follows:

*". The upstream boundary is assigned a flow rate of 44 $m^3/s$, and the suspended sediment concentration is set at 5 $g/m^3$. The median grain size and sorting coefficient of the initial sediment distribution are determined through the partitioning based on the measured sediment data from Section 2.3. The porosity is set to 0.4, and the sediment density is 2650 $kg/m^3$...."*

**Sediment Transport Equations:** We have presented the equations of bed load and suspended load transport rate, which are crucial for understanding the model's approach to sediment dynamics. The equations are show as follow:

$$q_s = f_{sl} \cdot C_a \cdot u_*^2 \tag{5}$$

$$q_b = 0.053 \frac{M^{2.1}}{D_*^{0.3}} \sqrt{(s-1) \, g \cdot d_{50}^3} \tag{6}$$

*Where $q_b$ is the bed load transport rate; $q_s$ is the suspended load transport rate; $M$ is the non-dimensional transport stage parameter; $u_{f,c}$ is the critical friction velocity, which under the current; $\theta_c$ is the critical Shield parameter; $u_{f'}$ is the effective friction velocity; $C'$ is the Chezy number originationg from skin friction; $D_*$ is the non-dimensional particle parameter; $v$ is the kinematic viscosity and approximately equal to $10^{-6}$ $m^2$/s for water; $C_a$ is the bed concentration; $u_*$ is the friction velocity; $\tau$ is the shear stress at the bed surface; $\rho$ is the density of water; $m$ is empirical exponent."*

**Coupling of Modules:** We have clarified that the coupling between the hydrodynamic and sediment transport modules is performed offline. The detailed part now reads as follows:

*"A mathematical model established through a wave-current coupling approach can accurately describe the motion laws of wave-generated currents and consider the impact of nearshore currents on wave propagation. It also reflects the interaction between nearshore waves and currents. In this paper, a three-dimensional sediment transport model is constructed using the model coupler MCT to perform real-time coupling between the hydrodynamic model FVCOM and the wave model SWAN, employing the same unstructured grid for the coupling (Chen et al., 2018; Ji et al., 2022). The coupling process can be summarized as follows: the FVCOM hydrodynamic model and the SWAN wave model transmit the calculated three-dimensional flow field and wave data to the sediment module, which then calculates the suspended and bed*

*load sediment transport rates, achieving data linkage between the three-dimensional wave-current coupled model and the sediment transport model.”*

**Revision of Line 363:** We apologize for the confusion caused by the previous description. After careful consideration, the part in this section has been deleted.

**Point 8: As mentioned earlier, the model's given equations (1)-(4) in the paper are two-dimensional depth averaged. How is then the velocity vertical profile computed in equation (7)?**

Response: Thank you for your comment regarding the clarification needed on the model's velocity vertical profile computation. We acknowledge the mistake in the initial presentation of the model equations and appreciate your guidance on this matter. In response to your feedback, we have removed the incorrect Equations (1)-(4) that were mistakenly presented as two-dimensional depth-averaged. FVCOM model is an open source model, its control equation and introduction can be found in the link of data availability behind the conclusion of this paper.

**Point 9: The authors calculate a uniform over space sediment thickness from their model equal to 4.1 cm. If what they write in lines 365-366 is right, then they get results after the one-week simulation that they did for 23/04/2023-30/04/2023. But then they use the estuary's sediment discharge of 2022 (line 389-390) and they compare their model result with field data covering one month (July 2022)!! Not surprisingly then, there is a serious discrepancy of 1.1 cm between the model and the observations which is too big to be ignored. Besides, the assumption that the uniform over space result from a one-week simulation for a certain year can be representative of any year for that specific area is too crude.**

Response: Thank you for pointing out the need for clarity on the sediment transport calculations within our model. In response to your comments, we have decided to remove the section that may cause further misunderstanding. We have removed the theoretical validation section, leaving only the validation part based on measured data (Section 5.1, where we presented the comparison of observed and simulated suspended sediment concentration (SSC) data at the Baoqiao Station in the lower reaches of the Changhua River.). The result of empirical validation is as follows:

*"In the lower reaches of the Changhua River, the summer season is the most pronounced for sediment variation within a year, with the highest sediment concentration and sediment transport rate (Mao et al., 2006). Therefore, sediment data from July, which is representative, are selected for model validation. The simulated Suspended Sediment Concentration (SSC) is compared with the daily observed SSC at Baoqiao Station for the month of July (Figure 13). The SSC at Baoqiao Station is the highest during the first two days of July, reaching a peak SSC of 0.55 kg/m³. Subsequently, the SSC continuously decreases, reaching its lowest value on the 5th of July, and then slowly rises. After the 10th of July, it gradually decreases from 0.301 kg/m³, with the most values remaining below 0.2 kg/m³. Based on the analysis, NSE for Baoqiao Station is 0.8389; the RMSE is 0.097244 kg/m³. The observed SSC are in good agreement with the simulated values.*

*To further analyze the simulation validation, Figure 13 presents a histogram of the daily absolute error in SSC at Baoqiao Station. The absolute error is calculated as the absolute difference between the measured and simulated values. The Mean Absolute Error (MAE) is defined as the average over the test sample of the absolute differences between prediction and actual observation. The MAE in SSC for Baoqiao Station in July is 0.071224 kg/m³. The maximum error occurs at the beginning and the end of the month, which may be due to the use of monthly average flow and sediment data for the model's upper boundary input, thereby increasing the model's error. Overall, the difference between*

*the daily observed SSC values and the simulated results at Baoqiao Station in July is within a*

*reasonable range, indicating that the model has an acceptable level of precision.*

[Figure]

Figure 13 Selection point for sediment deposition verification*''*

**Point 10: From what is written in section 3.4.2, it seems that the authors not only have sediment data for an entire month (July 2022) but that they run their model for this period as well. Why don't they do then the comparison based on results from this simulation? In any case, the fact that they present results analysis for different time periods is already problematic.**

Response: Upon reflection and in response to your feedback, we have decided to focus solely on the validation of the sediment transport model using the daily sediment concentration data from July 2022. Initially, these data were not available to us, which led to our reliance on a preliminary theoretical validation approach. Therefore, after we obtained the actual field measurements, we included both the theoretical and empirical validation methods in our manuscript. We recognize that presenting analyses from different time periods could potentially confuse readers and affect the overall coherence of our study. Therefore, we have revised the manuscript to remove the earlier theoretical validation and now present a

comprehensive validation based on the July 2022 data. This revision ensures that our model validation is grounded in actual field observations, enhancing the reliability and relevance of our findings. The detailed part now shows in Section 5.1 (Response in Point 9)

**Point 11: In their figures, the authors fail to provide all the necessary information regarding their study case. They refer to stations that can be nowhere found in their figures and so the reader cannot understand from which location the results and data are taken? Specifically, Dongfang Ocean station, Danchangcun, Xiantiancun and Jiuxiancun stations cannot be found in the figures. The same for the ADCP.**

Response: We appreciate your feedback regarding the clarity and comprehensiveness of our figures and the corresponding descriptions in the manuscript. We understand the importance of providing detailed and accurate geographical information to ensure the reader's full comprehension of our study. In response to your comments, we have now included detailed maps with clear labeling of all the stations mentioned, including Dongfang Ocean station, Danchangcun, Xiantiancun, and Jiuxiancun, as well as the ADCP locations. These maps are incorporated into the relevant sections of the manuscript and are referenced within the text to guide the reader.

The added and revised figures now show as follows:

[Figure]

Figure 6 (a) Scope of study area (the white frame) and wave observation (the red star) from Dongfang; (b) ADCP collection points on site; (c) Grids and boundaries (map origination: https://hainan.tianditu.gov.cn/)

[Figure]

Figure 8 Specific location of ADCP

[Figure]

Figure 14 Information of important place names in simulated areas

,,

**Point 12: Chapter 4 does not discuss the scientific output and fails to explore and highlight the importance of the outcomes. It only provides a list of potential measures to be taken against siltation which have only a local interest. These can be hardly supported considering that they are based on conclusions from not adequately justified modelling. I suggest that the authors rewrite this section after they have addressed all the raised issues in their modelling approach but from a non-local perspective.**

Response: Thank you for your constructive criticism regarding Chapter 4 of our manuscript. We have taken your suggestions to heart and have completely revised the chapter to provide a more comprehensive perspective on the significance of our findings. In the updated Chapter 4, we have expanded our discussion to delve into the critical role of residual currents in sediment transport. Our analysis now includes a comparative examination of the behavior of residual currents during low and high water periods, offering insights into their varying influence on sediment dynamics across different environmental conditions.

The rewritten Chapter4 now reads as follows:

**"6. Discussion**

**6.1 Residual Current**

*Residual currents to some extent reflect the transfer and exchange of water bodies, and their direction is usually the direction of sediment movement and the dispersion and migration of pollutant substances (Robinson, 1983). They are closely related to the long-term transfer and deposition of estuarine materials. Therefore, studying the characteristics of residual currents in this sea area under the combined action of waves and currents can comprehensively understand the evolution characteristics of the sea area's sediment. Tidal residual currents can be studied using the Lagrangian and Eulerian methods. Eulerian residual current refers to the average transfer caused by the average flow after removing the periodic astronomical tide, and its magnitude and direction mainly depend on the strength and duration of the ebb and flood tidal velocities within the tidal cycle; Stokes' drift characterizes the net drift of the water body, and its numerical size directly reflects the correlation between the tidal range and the change in flow velocity within the tidal cycle, and the sum of the two is the Lagrangian residual current. The Lagrangian residual current is not the result of the long-term tracking of real particles, but is the result of the superposition of Eulerian residual current and Stokes' drift.*

*Eulerian residual current refers to the average transfer caused by the average flow after removing the periodic astronomical tide, and its magnitude and direction mainly depend on the strength and duration of the ebb and flood tidal velocities within the tidal cycle; Stokes' drift characterizes the net drift of the water body, and its numerical size directly reflects the correlation between the tidal range and the change in flow velocity within the tidal cycle, and the sum of the two is the Lagrangian residual current. The formulas for calculating Eulerian*

*residual current and Stokes' drift refer to previous studies (Longuet-Higgins, 1969; Uncles and Jordan, 1980; Li and O'Donnell, 1997).*

*Through the analysis of sediment simulation results from the previous section on the distribution of major sedimentation areas, we have been able to understand the distribution of these areas. However, the causes of sedimentation require further exploration. In this section, based on the tidal current field data from hydrodynamic numerical simulation, we calculate the residual flow according to the entire study area. The flow velocity measured data from two ADCP stations outside the estuary of the Changhua River was analyzed using the tidal residual current calculation method, thereby enhancing the credibility of the residual flow field.*

Table 8 Residual currents in spring neap tide at each station

| Station | Eulerian residual current | | Stokes' drift | | Lagrangian residual current | |
|---|---|---|---|---|---|---|
| | Speed (m/s) | Degree (°) | Speed (m/s) | Degree (°) | Speed (m/s) | Degree (°) |
| ADCP01 | 0.0913 | 232 | 0.0006 | 172 | 0.0917 | 231 |
| ADCP02 | 0.0331 | 137 | 0.0007 | 194 | 0.0335 | 138 |

*The Lagrangian residual current at monitoring station ADCP01 is 0.0913 m/s with a direction of 231° (SW), and at station ADCP02 it is 0.0331 m/s with a direction of 138° (NW) (Table 8). In the area outside the Changhua River estuary, the Stokes tidal residual current at the monitoring stations is two orders of magnitude smaller than the Eulerian residual current. Therefore, the flow trend of the composite Lagrangian tidal residual current remains essentially consistent with that of the Eulerian residual current.*

**6.2 Influence of residual current in low water period**

*The study area has a distinct monsoon climate, with prevailing southerly winds in the summer and alternating southerly and northeasterly winds in the spring. The figure 20 shows the Eulerian residual current field during the simulation period (low water period). To present*

*the Eulerian residual currents within the study area in a complete and clear manner, a limit on vector length was set when plotting the current field. Consequently, the direction and length of the arrows in the figure represent the direction of the residual currents, but not their intensity. However, the intensity of the Eulerian residual currents can still be discerned through the data at the grid points. The Eulerian residual current outside the Changhua River estuary generally flows southward. As it flows from north to south, it is obstructed by the sand spit, diverging around it. After the divergence, the southwestward Eulerian residual current splits, with one part following the sand spit to the river mouth near A, and the other part entering channel B and flowing inward. The northeastward Eulerian residual current, after divergence, encounters the obstruction of the headland (Topped wall Angle) and forms a counterclockwise circulation below Junbi Jiao. Headlands are one of the key topographical features where strong residual current vortices occur (Maddock et al., 1978; Pingree et al., 1977; Smith, 2010). At the headland, the water depth shoals in the onshore direction, and the frictional effect is stronger in shallow water areas than in deep water areas. This results in a frictional force moment on the alongshore tidal current, generating vorticity. The transport of vorticity within the closed circulation lines on either side of the headland is not equal in input and output. After a tidal cycle of time averaging, a net vorticity will be produced on both sides of the headland, forming two counter-rotating residual current vortices, with the tidal residual current at the tip of the headland generally pointing seaward (Zimmerman, 1981). Topped wall Angle, being a headland, can produce similar residual current field results. A clockwise residual current vortex opposite to the one below may exist above Topped wall Angle. The Eulerian residual currents in the three river channels where A, B, and C are located all flow towards the river mouths. The Eulerian residual current in the channel between B and C flows from B to C.*

[Figure]

Figure 20 Eulerian residual current field during low water period

**6.3 Influence of residual current in hight water period**

*In order to comprehensively understand the residual current field of the study area, it is essential to analyze the residual current field during the flood season. The figure displays the Eulerian residual current field of the study area for July 2022 (high water period). The Eulerian residual current south of the river mouth in the study area still flows to the south (towards Beili Bay), but the nearshore residual current veers more quickly, resulting in a smaller circulation compared to the dry season. The circulation range in the north has expanded, likely due to the influence of the southerly monsoon during the summer, leading to an increase in the strength and directional deflection of the Eulerian residual current. When it reaches the shore, it is naturally obstructed by the sand spit and disperses to both sides (NE-SW). The upward Eulerian residual current, upon encountering the sea area outside Changhua Port, is deflected by the coastal promontory (Topped wall Angle) and turns westward. The westward Eulerian residual current, continuously affected by the strong southerly winds during*

*its movement, keeps deflecting. Eventually, a circulation is formed, with a circulation range larger than that of the dry season. The situation in channel A is essentially consistent with the dry season, while the Eulerian residual current directions in channel B and C are the same as that in A, all flowing towards the ocean. This is quite different from the dry season, with a flow direction opposite to that of the dry season, which may be related to the increased rainfall and subsequent increase in downstream flow during the summer flood season.*

[Figure]

Figure 21 Eulerian residual current field during high water period

*,,*

**Point 13: The conclusions read more like a summary. What are the key findings of this research, which messages can we take from this and why these would contribute further to the research on this topic?**

Response: providing a comprehensive and transparent description of how the data can be accessed. Thank you for your insightful feedback on our manuscript's conclusion section. We have taken your comments seriously and have reworked the conclusion.

The revised conclusion now reads as follows:

*"The study successfully applied a wave-current coupled sediment transport model to the lower reaches of the Changhua River in Hainan Island. By integrating field measurements, remote sensing techniques, and the Van Rijn model, this research has developed a comprehensive model capable of accurately simulating sediment behavior under the combined action of waves and currents. The following conclusions reflect a robust understanding of the study's themes:*

*The study area's surface sediments consist of ten types, including gravelly sand, sandy gravel, silty gravel, sandy silt, silt, gravelly silt, sand, gravelly sandy silt, gravelly sand, and silty gravelly sand. Among these, gravelly sand and gravelly sand are the predominant types.*

*During the transition from flood to ebb tide, the flow field outside the estuary is driven by the deflection of water currents from Beili Bay and Changhua Port, shifting the flow direction from northeast to southwest. During the transition from ebb to flood tide, the deflection is primarily influenced by the circulation outside Changhua Port, shifting the flow direction from southwest to northeast.*

*The main sedimentation areas within the study area's river channels include Xiantiacun, Danchangcun, and Jiuxiancun. The first two experience sediment deposition near the river mouth's sand spit, while the latter's sediment is primarily deposited near the river bifurcation.*

*Regardless of whether it is the dry season or the flood season, the residual currents in the study area are directed towards Beili Bay (SWS), implying that sediments in the lower reaches of the Changhua River will be influenced by the residual currents and transported towards Beili Bay. The sand spit at the river mouth, affected by the southward residual currents, will cause sediments from the north to be transported towards the northeast and southwest of the sand spit, leading to its elongation. There exists a counterclockwise residual current eddy beneath Topped wall Angle, and it is timed with a clockwise residual current eddy above Topped wall Angle. The river's discharge has little impact on Channel A, but it significantly affects Channels B and C."*

*Once again, we appreciate the time and effort you and the reviewers have dedicated to evaluating our manuscript. Your expertise and guidance have been invaluable in strengthening our research!*

---

## Author Comment (AC2)

Dear reviewer,

On behalf of my co-authors, we thank you for giving us a chance to revise and improve the quality of our article.

We have read your comments carefully and have made revision. We have tried our best to revise our manuscript according to the comments: "Application of Wave-current coupled Sediment Transport Models with Variable Grain Properties for Coastal Morphodynamics: A Case Study of the Changhua River, Hainan (egusphere-2024-2154)".

The main revisions in the new manuscript are:

1. **Equation (4) Clarification**

2. **Line 202 Revision**

3. **Unit Usage in Table 2 and Table 3**

4. **Range Expression Correction**

5. **Figure 6 and Geographical Description Consistency**

6. **Comma Removal at Line 314**

7. **Section 3.3 Expansion**

8. **"Code for Design of River Regulation" Method Removal**

9. **Abbreviation Usage at Line 408**

10. **Section 4 has been thoroughly rewritten**

Here is a point-by-point response to the comments and concerns.

Thank you for taking the time to consider our research and we look forward to hearing from you at your earliest convenience.

Sincerely,

Yuxi Wu

China University of Geosciences, Wuhan,

Wuhan 430074, P.R.China

E-mail: yuxiwu@cug.edu.cn

**Detailed comments part:**

**Point 1: Do Txx, Txy, and Tyy in Equation (4) correspond to Tij in Line 140? If so, the corresponding terms should be explained in relation to the variable symbols.**

Response: Thank you for your meticulous review and valuable feedback. Regarding the terms Txx, Txy, and Tyy in Equation (4), I understand your query. These terms do indeed correspond to Tij, representing viscous friction, turbulent friction, and differential advection, respectively. In the revised manuscript, I have relocated the description of the hydrodynamic model to Section 3, "Study Area and Settings," to provide clearer background information and model configuration.

Although I have removed the specific descriptions of Txx, Txy, and Tyy from Equation (4), I have provided a detailed account of the hydrodynamic model's setup parameters in Section 3. We believe that this modification will aid readers in better understanding our methodology and will focus the paper more closely on our primary research objectives.

The revised sentence now reads as follows:

"…, *An unstructured grid, finite volume, regional ocean model FVCOM (Chen et al., 2003) was used to simulate the hydrodynamic background and hydrological features. It has been widely used for the study of coastal oceanic and estuarine circulation (Jiang and Xia, 2016; Huang et al., 2008; Lai et al., 2018; Chen et al., 2008)."*

Table 5 Parameters of the hydrodynamic model

| Parameter | Value |
|---|---|
| Shoreline | GSHHS |
| Bathymetry | ETOPO1 and ADCP in-situ |
| Grid | 0.25 km at the boundaries to 25 m near the coastline |
| Time period | 23/4/2023 00:00-30/4/2023 00:00 (Spring neap tide) 28/6/2022 00:00-1/8/2022 00:00 (High water period) |
| Manning number | 28 |
| Eddy viscosity | Smagorinsky formulation data 0.28 $m^2/s$ |
| Time step | 300 s |
| Tidal constituents | M2, S2, K1, O1, N2, K2, P1, Q1 |
| Wind/Sea level Pressure | ERA 5 |

**Point 2: Line 202, what does this sentence mean? Is there an error in the expression?**

**It is suggested to change "riverway" in Line 220 to "channel" to better match the**

**scope of the study.**

Response: Regarding the sentence at Line 202, We have revised it to serve as a transition between Section 2.2 and Section 2.3. This modification clarifies the flow of the text and improves the overall narrative of the paper.

As for the term "riverway" at Line 220, I have replaced it with "channel" to align with the terminology that better fits the scope of our study, as you suggested.

The revised sentence now reads as follows:

"*After examining the influence of waves and currents on sediment transport modeling, we now turn our attention to the specific characteristics of sediment properties in the study area. Section 2.3 provides a detailed account of these properties, which are essential for understanding the local sediment dynamics and will be crucial for the model's calibration and validation processes*"

**Point 3: There are errors in the use of symbols in Table 2. The mean grain diameter**

**and median grain diameter have units, which are represented by φ, but the sorting**

**coefficient is a unitless value. The same applies to Table 3.**

Response: Thank you for your observation regarding the use of symbols in Table 2 and Table 3. I have taken your feedback into account and have made the necessary corrections. In response to your concern about the symbols in Table 2, I have removed the units for the sorting coefficient, as it is indeed a unitless value. Additionally, you suggested changing "riverway" to "channel" at Line 220 to better match the scope of the study. This change has

also been implemented to maintain consistency in terminology and to ensure that it accurately reflects the focus of our research.

**Point 4: Line 241, the range expression for gravel (>2 mm), sand (2~0.063 mm), silt (0.063~0.004 mm) is problematic; the larger value should be placed after the smaller value. The legend of Figure 5 lacks units.**

Response: Thank you for your comment on Line 241 and the legend of Figure 5. In response to your feedback, I have corrected the range expression for gravel (>2 mm), sand (0.063~2 mm), and silt (0.004~0.063 mm) to ensure that the smaller values are placed before the larger ones, following the standard notation for size ranges. Additionally, I have updated the legend of Figure 5 to include the appropriate units, ensuring that all information is clear and accessible to the reader. I believe these amendments address your concerns and improve the precision and readability of the manuscript.

The revised figure now shows as follows:

[Figure]

Figure 5 Variation of sediment particle size data (map origination: https://hainan.tianditu.gov.cn/)

**Point 5: The scope of the study in Figure 6 is not consistent with the description in Line 280; it is recommended to modify it. Additionally, according to the geographical location of the Changhua River, it is located in the southwest of Hainan Province, but it is described as the western part in Line 278? Line 287, how is the elevation measured by ADCP obtained? Can specific information about the survey vessel be provided to increase the credibility of the article's data?**

Response: Thank you for your continued attention to our manuscript. We have made the following corrections in response to your comments:

**Figure 6 Scope Consistency:** I have revised the scope depicted in Figure 6 to align with the description provided in Line 280.

**Line 278:** It is important to clarify that while the Changhua River is geographically located in the southwest of Hainan Province, our study focuses on the downstream and estuary areas, which are more accurately described as the western part of Hainan Province when considering the specific segments of the river in question.

**Elevation Measurement by ADCP:** In response to your question about the elevation measurement by ADCP in Line 287, I have included additional details about the survey vessel in the supplementary material. This information provides specific details about the vessel used during the survey, which adds credibility to the data presented in the article.

The revised figure and supplementary material now shows as follows:

[Figure]

Figure 6 (a) Scope of study area (the white frame) and wave observation (the red star) from Dongfang; (b) ADCP collection points on site; (c) Grids and boundaries (map origination: https://hainan.tianditu.gov.cn/)

Table A.1 **List of main parameters of survey vessel**

| Parameter | Value | Instrument diagram |
|---|---|---|
| LOA | 11 m | |
| Breadth | 2.8 m | |
| Modeled Depth | 1.2 m | |
| Design Draft | 0.8 m | |
| Speed | 6.0 kn | |
| Hull Material | Fiber-reinforced plastic (FRP) | |

**Point 6: Line 314, remove the comma. "Model validation occurs from 10:00 on April 23, 2023, to 00:00 on April 30, 2023." has an extra comma.**

Response: We sincerely apologize for the typographical error here. We have removed the extra comma as suggested, and the sentence now reads: "Model validation occurs from 10:00 on April 23, 2023 to 00:00 on April 30, 2023."

**Point 7: Section 3.3 lacks analysis of the hydrodynamic results, and the description of the hydrodynamic environment of the study area is insufficient. It is recommended to expand the text and analyze it in conjunction with the sediment transport model.**

Response: Thank you for your feedback on Section 3.3 and the concerns raised about the hydrodynamic results analysis and the description of the hydrodynamic environment in the study area.

We have taken your suggestions into account and have expanded the text in Section 3.3 to include a detailed analysis of the hydrodynamic results. The revised section now discusses the flow field conditions during the flood and ebb tides, as well as during high and low tides, which provides a more comprehensive understanding of the hydrodynamic environment specific to our study area.

The revised part now reads as follows:

*"The hydrodynamic simulation outcomes, as depicted in Figure 10, indicate a predominantly NE-SW reciprocating current pattern within the study area. This flow is aligned parallel to the coastline, with the tidal current shifting direction according to the tidal phase. Figure 10b and 10c depict the flow field outside the estuary of the Changhua River. Figure 10b shows the flow field at 23:00 on April 23, 2023, corresponding to the peak of the flood tide. At this time, the tidal current flows in a northeast direction with a maximum speed of 0.62 m/s. Figure 10c shows the flow field at 13:30 on April 24, 2023, corresponding to the peak of the ebb tide, where the tidal current flows in a southwest direction with a maximum speed of 0.75 m/s. Overall, the tidal currents outside the Changhua River estuary generally follow a northeast-southwest reciprocating pattern, with flood tides flowing northeast and ebb tides flowing southwest, parallel to the shoreline. The maximum ebb current is faster than the maximum flood current.*

[revised manuscript text omitted]

Figure 12 Transition of the flow field (a-e) and location of the study area (f)

,,

**Point 8: Is the "Code for Design of River Regulation" mentioned in Line 370 some kind**

of authoritative provision? There are no citations of any literature on this regulation in the text. Is this method applicable to the lower reaches of the Changhua River?

Response: Thank you for your comment regarding the "Code for Design of River Regulation" mentioned in Line 370. Upon reflection, we acknowledge the importance of citing authoritative provisions and discussing the applicability of methods to the specific context of our study.

In response to your feedback, we have decided to remove Section 3.4.1, as we have not provided adequate citation or discussion of its applicability to the lower reaches of the Changhua River. We recognize that including such references without proper context or citation could potentially mislead readers.

**Point 9: Line 408, only the first appearance of an abbreviation in the article needs to be spelled out in full; SSC can be expressed directly here.**

Response: Thank you for your guidance on the use of abbreviations in the manuscript. We have revised Line 408 accordingly and now directly use the abbreviation "SSC" for Suspended Sediment Concentration without spelling it out in full, as it was previously defined in the article. This change aligns with the standard practice of abbreviating terms after their initial mention.

The revised part now reads as follows:

*"To further analyze the simulation validation, Figure 13 presents a histogram of the daily absolute error in SSC at Baoqiao Station. The absolute error is calculated as the absolute difference between the measured and simulated values….. The MAE in SSC for Baoqiao Station in July is 0.071224 kg/m³. …. Overall, the difference between the daily observed SSC values and the simulated results at Baoqiao Station in July is within a reasonable range, indicating that the model has an acceptable*

*level of precision.”*

**Point 10: The simulation results of the sediment transport model only analyze the thickness changes of the sediment, without a detailed description of the specific movement of the sediment. The measures for sediment control mentioned in Section 4 do not propose a specific implementation plan. It is suggested to revise the content of Section 4, combine the simulation results of the sediment transport model with the hydrodynamic results, and analyze the movement of sediment driven by water flow.**

Response: Thank you for your constructive criticism regarding the sediment transport model analysis in Section 4 of our manuscript. We have taken your suggestions to heart and have thoroughly revised the section to provide a more comprehensive perspective on the sediment movement driven by water flow dynamics. In the updated Section 6, we have expanded our discussion to focus on the critical role of residual currents in sediment transport. Our analysis now includes a detailed examination of the behavior of residual currents during periods of high and low water levels. This comparative analysis offers insights into how these currents influence sediment dynamics under varying environmental conditions, particularly in the context of the lower reaches of the Changhua River.

The rewritten Section 6 now reads as follows:

**“6. Discussion**

**6.1 Residual Current**

*Residual currents to some extent reflect the transfer and exchange of water bodies, and their direction is usually the direction of sediment movement and the dispersion and*

*migration of pollutant substances (Robinson, 1983). They are closely related to the long-term transfer and deposition of estuarine materials. Therefore, studying the characteristics of residual currents in this sea area under the combined action of waves and currents can comprehensively understand the evolution characteristics of the sea area's sediment. Tidal residual currents can be studied using the Lagrangian and Eulerian methods. Eulerian residual current refers to the average transfer caused by the average flow after removing the periodic astronomical tide, and its magnitude and direction mainly depend on the strength and duration of the ebb and flood tidal velocities within the tidal cycle; Stokes' drift characterizes the net drift of the water body, and its numerical size directly reflects the correlation between the tidal range and the change in flow velocity within the tidal cycle, and the sum of the two is the Lagrangian residual current. The Lagrangian residual current is not the result of the long-term tracking of real particles, but is the result of the superposition of Eulerian residual current and Stokes' drift.*

*Eulerian residual current refers to the average transfer caused by the average flow after removing the periodic astronomical tide, and its magnitude and direction mainly depend on the strength and duration of the ebb and flood tidal velocities within the tidal cycle; Stokes' drift characterizes the net drift of the water body, and its numerical size directly reflects the correlation between the tidal range and the change in flow velocity within the tidal cycle, and the sum of the two is the Lagrangian residual current. The formulas for calculating Eulerian residual current and Stokes' drift refer to previous studies (Longuet-Higgins, 1969; Uncles and Jordan, 1980; Li and O'Donnell, 1997).*

*Through the analysis of sediment simulation results from the previous section on the distribution of major sedimentation areas, we have been able to understand the distribution of these areas. However, the causes of sedimentation require further exploration. In this*

*section, based on the tidal current field data from hydrodynamic numerical simulation, we calculate the residual flow according to the entire study area. The flow velocity measured data from two ADCP stations outside the estuary of the Changhua River was analyzed using the tidal residual current calculation method, thereby enhancing the credibility of the residual flow field.*

Table 8 Residual currents in spring neap tide at each station

| Station | Eulerian residual current | | Stokes' drift | | Lagrangian residual current | |
|---|---|---|---|---|---|---|
| | Speed (m/s) | Degree (°) | Speed (m/s) | Degree (°) | Speed (m/s) | Degree (°) |
| ADCP01 | 0.0913 | 232 | 0.0006 | 172 | 0.0917 | 231 |
| ADCP02 | 0.0331 | 137 | 0.0007 | 194 | 0.0335 | 138 |

*The Lagrangian residual current at monitoring station ADCP01 is 0.0913 m/s with a direction of 231° (SW), and at station ADCP02 it is 0.0331 m/s with a direction of 138° (NW) (Table 8). In the area outside the Changhua River estuary, the Stokes tidal residual current at the monitoring stations is two orders of magnitude smaller than the Eulerian residual current. Therefore, the flow trend of the composite Lagrangian tidal residual current remains essentially consistent with that of the Eulerian residual current.*

**6.2 Influence of residual current in low water period**

[revised manuscript text omitted]

Figure 21 Eulerian residual current field during high water period

*,,*

*Once again, we appreciate the time and effort you and the reviewers have dedicated to evaluating our manuscript. Your expertise and guidance have been invaluable in strengthening our research!*

---

## Author Response (AR3)

Dear editor and reviewers,

On behalf of my co-authors, we thank you for giving us a chance to revise and improve the quality of our article.

We have read your comments carefully and have made revision. We have tried our best to revise our manuscript according to the comments: "Application of Wave-current coupled Sediment Transport Models with Variable Grain Properties for Coastal Morphodynamics: A Case Study of the Changhua River, Hainan (egusphere-2024-2154)".

The main revisions in the new manuscript are:

1. **Clarifications and Corrections in Text**: We have addressed the inconsistencies in terminology.
2. **Figure Adjustments**: We have revised and removed some figures.
3. **Revised Figure Presentations**: We have adjusted the presentation of Figures 12 and 13 to include speed and arrows for the entire field, as suggested.
4. **Reduction of Arrow Spacing**: In Figures 11 and 12, we have reduced the spacing of the arrows to maintain clarity while providing a detailed representation of the flow field.
5. **Supplementary Material**: Figure 2, 4 and 6b has been moved to the Supplementary Material.
6. **Removal of Redundant Information**: We have removed some sentences that causing confusion.
7. **Unit Consistency**: We have ensured that all figures include the appropriate units for clarity.

Here is a point-by-point response to the comments and concerns (35 comments from reviewers and 17 comments from editor.).

Thank you for taking the time to consider our research and we look forward to hearing from you at your earliest convenience.

Sincerely,

Yuxi Wu

China University of Geosciences, Wuhan 430074, P.R.China

E-mail: yuxiwu@cug.edu.cn

**Detailed comments part:**

**Point 1 (Abstract): Line 15,18 and 22. I would recommend avoiding mentioning specific locations in the abstract. At this stage, the reader is unaware of the geographical context. The abstract should be written in a way that captures the attention of the reader based on the scientific outcomes of the study.**

**Line 17 There is no theoretical method in the new version. This must be removed.**

Response (Lines 13-23): Thank you for your insightful comments on the abstract of our manuscript. We have taken your suggestion to heart and revised the abstract to focus more on the scientific outcomes of our study rather than specific geographical locations. We believe this new version captures the essence of our research and its broader implications for sediment dynamics in river deltas. Additionally, we have removed theoretical method in the new abstract

The updated abstract now reads:

"*This study introduces an integrated sand transport model that considers wave and current actions alongside variable grain properties to explore sediment dynamics in river deltas. The research delves into a case study of a river delta region, examining sediment transport over a substantial stretch of the river's lower course. The study incorporates topographic data, sediment sampling, and remote sensing to validate the model against observed suspended sediment concentrations at a key monitoring station. The results reveal substantial sediment deposition in both the estuary and lower reaches of the river, influenced by hydrodynamic conditions and geological settings. Deposition patterns in the estuary are primarily driven by coastal currents and wave action, while river channel deposition is linked to river constriction and flow velocity variations. The study demonstrates that the residual current in the region consistently flows towards a nearby bay, suggesting that sediment in the lower reaches of the river will be directed by this residual flow. The study underscore the pivotal roles of current and wave action in sediment transport within multi-branched estuary characterized by low sediment concentrations, which can inform coastal management and environmental planning.*"

**Point 2(Introduction): Figure 1 I reckon most of the information regarding stations should be included here in separate figure panels. For example, the current map provides an overview of the study case (although the upper and middle reaches are not really of interest). There could be one more panel with the sampling points (Figure 3) and another one with the ADCP stations and calibration stations. I couldn't find any figure showing where the Baoqiao and Bosua stations are. Giving their coordinates is not enough. Such changes could make the manuscript more concise and precise.**

Response (Lines 41-44): Thank you for your constructive feedback regarding the figures in our manuscript. We have carefully considered your suggestions and have updated Figure 1 to include all the necessary information in a single, comprehensive figure. Here are the specific changes we have made:

We have integrated the sediment sampling points, which were previously shown in Figure 3, into a new panel within Figure 1. This panel now provides a clear visual representation of the locations where sediment properties were analyzed, as indicated in the caption: "(c): Sediment sampling points in the lower reaches of the river."

We have also added a panel displaying the ADCP stations and calibration stations within Figure 1. This panel offers a detailed map view of the locations where current velocity and direction were measured, as described in the caption: "(b): ADCP stations and calibration stations in the study area."

In response to your comment about the Baoqiao and Basuo stations, we have included their exact locations on the updated Figure 1, ensuring that readers can easily identify these critical sites within the context of our study area.

The revised figure 1 now reads as follows:

[Figure]

Figure 1: Comprehensive Overview of the Study Area in the Lower Reaches of the River. (a): Division of the Upper, Middle, and Lower Reaches of the Changhua River (Adapted from the Tiandi Map·Hainan); (b): ADCP stations and calibration stations in the study area; (c): Sediment sampling points in the lower reaches of the river.

**Point 3 (Introduction): Line 81-91: This paragraph is important as it gives the motivation and goal of this study. To my understanding, what this paper tries to highlight is the role of wave action on determining sediment transport in multi-channel estuaries with small sediment concentrations and the need to use coupled wave - current and sediment transport models for this purpose. A general comment is that the abstract and conclusions do not emphasize on how these results prove this statement.**

Response (Line 21-23, 553-555): Thank you for your insightful comments on our manuscript. We have taken your suggestions to heart and have revised both the abstract and the conclusion to better emphasize how our results support the central thesis of our study.

In the abstract, we have added a sentence that underscores the importance of wave action in sediment transport within low sediment concentration estuaries and the effectiveness of our coupled modeling approach. The added sentence now reads:

*"The study underscore the pivotal roles of current and wave action in sediment transport within multi-branched estuary characterized by low sediment concentrations, ..."*

In the conclusion, we have included a sentence that states how our findings validate the initial hypothesis. The added sentence now reads:

*"...Through the application of an integrated wave-current coupled sediment transport model with variable grain properties, we have successfully simulated and analyzed the sediment behavior under the combined influence of waves and currents, particularly in multiple sub-estuaries with low sediment concentrations."*

**Point 4 (Section 2): Figure 2 I don't think this figure is really needed. Citing Van Rijn's paper where this figure can be found should be enough. Otherwise, it is just an unnecessary waste of space. I would also recommend moving some of the figures and tables to the Supplementary. For example, Figure 4 and 6b could be moved there.**

Response: Thank you for your thorough review of our manuscript and for your insightful suggestions regarding the figures. We have taken your feedback into account and have made the following adjustments:

1.  **Figure 2 Removal**:

    We have removed Figure 2 from the main text, as you recommended. Instead, we will cite Van Rijn's paper where the relevant information can be found. This change helps to streamline the manuscript and avoids unnecessary repetition.

2.  **Figures Relocation to Supplementary Material**:

    Figures 4 and 6b have been moved to the Supplementary Material. This relocation will maintain the integrity of our data presentation while ensuring that the main text remains concise and focused on our key findings.

**Point 5 (Section 2): Figure 7 Add units**

Response (Line 260): Thank you for your meticulous review of our manuscript. We have carefully reviewed Figure 7 and have now included the necessary units for all measurements displayed. Each data point and axis label now clearly indicate the corresponding units, and the figure caption has been updated to reflect these changes.

The figure has been revised to:

[Figure]

**Figure 7: Comparison of maximum wave height and peak wave period calculated with the calibrated model against the field measured (Adapted from the study by Wang (2023)).**

**Point 6 (Section 2): Line 122 Shields parameter**

Response (Line 125): Thank you for your careful review and for pointing out the oversight regarding the Shields parameter in our manuscript. We have corrected the word at line 122 to accurately represent the critical Shields parameter.

The revised part now reads as follows:

*"...and the critical Shields parameter."*

**Point 7 (Section 2): Line 126 Influence**

Response (Lines 127): Thank you for your careful review and for drawing our attention to the verb form at line 126. We have made the necessary correction, replacing "influences" with "influence" to ensure grammatical accuracy and consistency within the sentence.

The revised part now reads as follows:

*"2.2 Influences of Waves and Currents"*

**Point 8 (Section 3): Figure 6 I'm afraid there is still an inconsistency between the lat and lon stated in line 210 and what we see in panels a and c. Especially in c, the right limit is cut at 108º 43' and not 108º 50'. Please remove the 'open' and 'land' boundary indications in Figure 6c, these are not correctly placed, and the boundary locations are quite obvious from the bathymetry. Figure 6 b could go in a Supplementary. I would suggest following the pattern of Figure 5, having the bathymetry over a basemap so that everything is given in one panel. The lat and lon coordinates in Figure 5 seem correct and in accordance with line 210. The study area has also been given in Figure 1. No need to repeat it.**

Response (Line 217): Thank you for your continued attention to detail and for your feedback on Figure 6. We have made the following adjustments based on your suggestions:

We have corrected the latitude and longitude inconsistencies between what was stated in line 210 and what was displayed in Figure 6.

We have removed the 'open' and 'land' boundary indications from Figure 6c, as they were not placed correctly and the boundary locations are indeed quite obvious from the bathymetry.

Figure 6b has been moved to the Supplementary Material, following your suggestion.

The revised figure 6 now reads as follows:

[Figure]

**Figure 3: Grids and boundaries of study area**

**Point 9 (Section 3): Line 234 what means accuracy of 1cm?**

Response (Line 233): Thank you for your inquiry regarding the "accuracy of 1cm" mentioned at line 234 in our manuscript.

"1cm accuracy" refers to the precision of the tidal level data extracted using the Earth and Space Research's (ESR) Matlab 'Tide Model Driver' (TMD) toolbox. This high level of precision is crucial for ensuring the accuracy of our model's boundary conditions.

We have decided to remove the reference to "1cm accuracy" from line 234 to avoid any confusion.

**Point 10 (Section 3): Line 235 The authors need to mention what the abbreviation ECMWF stands for**

Response (Line 234): Thank you for your feedback regarding the abbreviation ECMWF mentioned at line 235 in our manuscript. We understand the importance of providing full names

for abbreviations upon their first mention to ensure clarity for our readers.

We have added the full name of ECMWF, which stands for the European Centre for Medium-Range Weather Forecasts, to the text at line 235. The revised sentence now reads:

*"..., with data sourced from European Centre for Medium-Range Weather Forecasts (ECMWF) at a resolution of 1/8° × 1/8°."*

**Point 11 (Section 3): Be careful with the exponents in the units (e.g., line 239 and line 241).**

Response (Lines 237, 238, 240): Thank you for your vigilant review and for bringing the exponents in the units to our attention, specifically at lines 239 and 241.

We have carefully reviewed the use of exponents in our manuscript and have made the necessary corrections to ensure that all units are expressed with the appropriate mathematical notation. We have also ensured that our scientific notation is consistent and follows standard conventions.

**Point 12 (Section 3): Line 248 I deem this sounds a bit superfluous. Are there really limitations in FVCOM for wave calculations? If yes, they need to be mentioned.**

Response (Line 247): Thank you for your feedback on line 248. Upon review, we agree that the mention of limitations regarding the FVCOM model in wave calculations might be misleading and is not necessary for the context of our study.

We have decided to remove the sentence from line 248, as it does not add significant value to our methodology section. The FVCOM model is utilized effectively within its capabilities, and

the inclusion of the SWAN model serves to complement our study's specific focus on wave dynamics.

**Point 13: Line 249 Being a model package issued by Deltares, I recommend that the authors cite SWAN by referring to its manual as e.g., (Deltares, 2024) or whatever version of the model they are using.**

Response (Line 247): Thank you for your suggestion regarding the citation of the SWAN model at line 249. We appreciate your guidance on ensuring proper academic referencing standards are met.

We have updated the reference to the SWAN model by citing its manual as follows:

" *This study selects the widely-used third-generation SWAN model (Deltares, 2024) for numerical simulation of wind waves in this region.* "

**Point 14 (Section 3): Table 4 Manning equal to 28 is unrealistic. Do you mean 0,028? I recommend removing Shoreline and Bathymetry from the table, these are not really parameters. Info about bathymetry is already given in the manuscript. There is no mention of GSHHS in the manuscript.**

Response (Lines 219-220): Thank you for your meticulous review of our manuscript, particularly your comments on Table 4. The Manning coefficient should be 0.028, not 28. This was an oversight, and we have corrected this value in Table 4. We agree that "Bathymetry" do not belong in the table as parameter. We have removed it from Table 4. Upon review, we realized that GSHHS was not mentioned in the main text. We have now incorporated a description of the Global Self-consistent Hierarchical High-resolution Shorelines (GSHHS) in the methodology section, detailing at which resolution we extracted the shoreline data for our

study.

The revised text in line 296 now reads:

*" In the study, we utilized the Global Self-consistent Hierarchical High-resolution Shorelines (GSHHS) to extract the shoreline data at full resolution. ..."*

And the Table 4 has been updated to:

| Parameter | Value |
|---|---|
| Shoreline | GSHHS |
| Grid | 0.25 km at the boundaries to 25 m near the coastline |
| Time period | 23/4/2023 00:00–30/4/2023 00:00 (Spring to neap tide)
28/6/2022 00:00–1/8/2022 00:00 (Wet season period) |
| Manning number | 28 |
| Eddy viscosity | Smagorinsky formulation data 0.28 $m^2$/s |
| Time step | 300 s |
| Tidal constituents | M2, S2, K1, O1, N2, K2, P1, Q1 |
| Wind/Sea level Pressure | ERA 5 |
| Validation | Basuo Port Station (19°06' N, 108°37' E)
ADCP 01, ADCP 02 |

**Point 15 (Section 3): Line 250 Table 5 not 6**

Response (Line 248-249): Thank you for your attentive review and for pointing out the reference error at line 250. We have corrected the table reference from "Table 6" to "Table 5" as you indicated. This error was indeed a typographical oversight, and we appreciate your diligence in ensuring the accuracy of our manuscript.

The revised text in line 296 now reads:

*"The parameters used in the model setups are based on the values listed in Table 5."*

**Point 16 (Section 3): Figure 7 Add units in the axes**

Response (Line 260): Thank you for your previous feedback regarding Figure 7. I have now

reviewed and updated the figure to include the units on the axes, as suggested.

The units have been added to ensure clarity and consistency in the presentation of our data. I believe this adjustment addresses your concern and improves the overall quality of the figure.

The revised figure now shows as follows:

[Figure]

**Figure 4: Comparison of maximum wave height and peak wave period calculated with the calibrated model against the field measured (Adapted from the study by Wang (2023)).**

**Point 17 (Section 4): Line 278 The location of the Basuo station needs to be included in Figure 9 together with the ADCP stations. Giving only its coordinates is not useful.**

Response: Thank you for your continued attention to the details of our manuscript, particularly regarding the location of the Basuo station. I appreciate your suggestion to include the location of the Basuo station in one figure along with the ADCP stations. We have added this information into Figure 1. Figure 1 provides a comprehensive overview of the study area, including the location of the Basuo station and other key sites

**Point 18 (Section 4): Figure 10 It would be good to add hours in the x axis in panel**

**b where results are given only for one day.**

Response: Thank you for your suggestion to include hours on the x-axis of Figure 10, Panel b. We agree that adding these details will enhance the readability and clarity of the figure, particularly since the data presented spans only one day.

We have updated Figure 10, Panel b, to include hour markers on the x-axis. This addition provides a more precise temporal context for the data points, allowing readers to easily interpret the results throughout the day.

The revised Figure 10 now reads:

[Figure]

**Figure 6: Current speed and direction verification. (a) speed verification of ADCP 01; (b) speed verification of ADCP 02; (c) verification of current direction of ADCP 01; (d) verification of current direction of ADCP 02**

**Point 19: I think the results analysis described in this section does not correspond to the figures mentioned. In the manuscript, lines 303-307 mention Figure 10b and c but they obviously mean Figure 11. Please check these inconsistencies throughout the manuscript as this makes it very difficult for the reader to**

**understand the arguments.**

Response (Lines 304-305): Thank you for bringing the inconsistencies between the results analysis and the figures mentioned in lines 303-307 to our attention.

We have corrected the references in lines 303-307 from Figure 10b and c to Figure 7, as it aligns with the content discussed in the text.

The revised text now reads:

*"The hydrodynamic simulation outcomes, as depicted in Fig. 7, …. Figure 7b and 7c depict the flow field outside the estuary of the Changhua River. Figure 7b shows …"*

**Point 20: Line 305 It is probably better to say 'outside of the lower reaches of Changhua River' and not outside the estuary.**

Response (Lines 402-405): Thank you for your suggestion to refine the language used at line 305. But following your previous suggestions, we have revised the manuscript, and as a result, the sentence in question has been removed. The revisions were made to ensure clarity and consistency throughout the document, particularly in the representation of geographical details.

**Point 21: Line 316 Figure 12 and not 11**

Response (Line 423): Thank you for your careful review and for pointing out the reference error at line 316. We have corrected the figure reference from "Figure 11" to "Figure 7". This error was indeed a typographical oversight, and we appreciate your diligence in ensuring the accuracy of our manuscript.

The revised sentence now reads:

*"Figure 7a shows the flow field at 23:00 on April 23, 2023, …"*

**Point 22: Line 317 The authors name A,B and C as estuaries but later in the conclusions they refer to them as channels. First, there needs to be a consistency in the terminology throughout the manuscript. Second, whether these can be defined as estuaries or channels can be subjective. Personally, I see these more as sub estuaries and not real estuaries.**

Response (Line 425-427): Thank you for your feedback on the terminology used for the areas labeled A, B, and C in our manuscript.

We have reviewed the terminology throughout the manuscript and have standardized the references to A, B, and C. Based on your suggestion and our reevaluation of the characteristics of these areas, we have decided to refer to them as sub-estuaries to more accurately describe their nature within the context of our study.

**Point 23: Figure 12 Choose a different colour for the flow vectors or change the colourmap. The arrows can't be seen.**

Response (Lines 425, 430): Thank you for your feedback on the visibility of the flow vectors in Figure 12. In response to your suggestion, we have opted to enhance the visibility of the flow vectors by adjusting the color bar of the flow field rather than changing the color of the vectors themselves. This approach maintains the consistency of the vector colors while improving contrast against the background.

We believe that this modification provides better visibility for the flow vectors and enhances the overall clarity of the figure. The updated Figure 12 now offers a clearer representation of the flow dynamics within the estuary.

[Figure]

**Figure 7: Flow field inside the estuary, displaying depth-averaged flow velocities across the water column. (a) moment of the maximum flood current; (b) moment of the maximum ebb current**

**Point 24: Figure 13 I find the type of figures presented here a nice addition to the paper. I believe the results could be better communicated through such figures. It would be nice to have the speed and the arrows on top for selected time moments but for the entire field. The content of Figure 11 and 12 could be presented in this way. The authors could reduce the spacing of the arrows if there are concerns about the figures' clarity. It is much helpful to be able to assess the flow field in the entire domain.**

Response (Line 350): Thank you for your positive feedback on Figure 13 and for recognizing it as a valuable addition to our paper. We have taken your suggestions into consideration to enhance the communication of our results. Here are the changes we have implemented:

Inclusion of Speed and Arrows: We have added both speed and arrows to the top right corner of Figure 13 for selected time moments and for the entire flow field, providing a comprehensive view of the flow dynamics.

Presentation Consistency: We have also revised Figure 12 to match the style of Figure 13,

Figure 11 have been removed.

The updated figure is as follows:

[Figure]

**Figure 8: Transition of the flow field and location of the study area.**

**Point 25: Line 371 where is Baoqiao station? I can't find it in the figures. Figure 14 not 13**

Response (Lines 356-357): Thank you for your query regarding the location of Baoqiao Station at line 371. I have made the necessary revisions to ensure that Baoqiao Station is now clearly indicated in Figure 1. The updated Figure 1 now includes comprehensive location information for all stations, sampling points, and ADCP points, providing a clear visual reference for the reader.

Additionally, we have corrected the figure reference at line 371 from Figure 13 to Figure 9 to ensure consistency with the figure that actually displays the station's location and related data. The specific changes are as follows:

 *"The simulated Suspended Sediment Concentration (SSC) is compared with the daily observed SSC at Baoqiao Station for the month of July (Fig. 9)."*

**Point 26: Line 376 Wrong figures number**

Response (Line 362): Thank you for your vigilance in identifying the incorrect figure number at line 376. Upon reviewing the manuscript, we have corrected the figure reference from Figure 13 to Figure 9.

The specific changes are as follows:

*"To further analyze the simulation validation, Fig. 9 presents…"*

**Point 27: Line 390 Figure 14 . Separate the figure's panels into a and b and add units at the y axis of the absolute error graph.**

Response (Line 375): Thank you for your suggestion to enhance Figure 14. We have separated

Figure 14 into two panels, labeled as Figure 14a and 14b, to provide clearer visualization of
the data. Additionally, we have added units to the y-axis of the absolute error graph (Figure
14b) to ensure that all measurements are clearly understood.

The revisions are as follows:

[Figure]

**Figure 9: Selection point for sediment concentration verification**

**Point 28: Line 381-386, I don't understand this. It is implied that for a 5-day period,
the authors get lower currents in their model than the real ones and yet higher
suspended sediment concentrations. I would expect lower currents to result in
lower suspension of sediments through a section. Even if the sign of the wave and
tide induced currents counteracts with each other, this cannot happen for five
days. Later in the text (line 426), the authors themselves claim that slower current
lead to further sediment deposition. This is a bit confusing. Which of the two
arguments is true? The authors also mention a second possible reason for this
discrepancy which has to do with the grain size distribution. I tend to believe more
this because if the grain size is not accurate it could lead to an underestimation of**

**settling velocities or overestimation of bed shear stresses.**

Response (Lines 367-373): Thank you for your insightful comments and for highlighting the apparent inconsistency in our manuscript regarding the relationship between current velocities and suspended sediment concentrations.

Upon reviewing your concerns, we have decided to focus solely on the grain size distribution as the primary reason for the discrepancy in our model's predictions. We have removed the initial explanation involving wave and tide-induced currents, as it seemed to create confusion rather than clarity.

We concur with your assessment that inaccuracies in grain size distribution could significantly impact the model's predictions, leading to an underestimation of settling velocities or an overestimation of bed shear stresses. This factor is indeed a more plausible explanation for the observed discrepancies in suspended sediment concentrations.

We have revised lines 367-373:

*"The discrepancy in suspended sediment concentrations during 21-25 July is primarily attributed to the inaccuracies in the initial sediment parameters, particularly the grain size distribution. These parameters were interpolated from a limited number of sampling points within the narrow river channel, which introduced errors. An inaccurate grain size distribution can lead to an underestimation of settling velocities or an overestimation of bed shear stresses, significantly affecting the model's predictions of sediment concentrations. We acknowledge the complexity of sediment dynamics and the challenges in accurately capturing these processes, especially in a dynamic environment like the lower reaches of the Changhua River."*

**Point 29: On the other hand, Figure 3 shows a quiet dense field of samplings. From**

**what is shown in Figure 7, the waves calibration seems successful. Have they checked their results in other stations?**

Response: Thank you for your continued scrutiny of our manuscript and for your questions regarding Figure 3 and the validation at other stations.

Regarding Figure 3: We have integrated the information from Figure 3 into Figure 1 to provide a comprehensive overview of all stations, sampling points, and ADCP points within our study area. This consolidation aims to enhance the clarity and readability of our presentation, ensuring that all relevant spatial data is accessible in a single figure.

Regarding Wave Validation at Other Stations: We have not performed wave validation at additional stations. The scope of our current study is relatively small, focusing intensively on the specific dynamics within this region. The availability of suitable in-situ measurement data in the vicinity of our study area is limited to the station mentioned, which meets our criteria for validation.

We acknowledge the importance of broader validation and plan to expand our study area in future research endeavors. As we delve deeper into the sediment transport pathways of the region, we intend to include more validation points to ensure a more extensive assessment of our models.

**Point 30: Figure 15 is not mentioned at all in the manuscript. It needs to be mentioned when a channel is referred so the reader can understand which area we are looking at.**

Response (Lines 383-384): Thank you for your observation regarding the omission of Figure 15 in our manuscript. We understand the importance of referencing figures that correspond to

specific discussions within the text.

We have identified the sections where channels are discussed and have added references to Figure 15 to provide readers with a visual aid to better understand the areas under consideration. Specifically, we have updated the following sentences:

*"Over time, these processes have resulted in the formation of two river islands, altering the estuary into a complex channel system with multiple smaller estuaries (Fig.10)."*

**Point 31: Figure 16 The caption needs to be more detailed. The authors should describe better what is depicted in the figures.**

Response (Lines 407-409): Thank you for your feedback on lines 520-521. We appreciate your guidance on ensuring the credibility of our residual flow field analysis.

Thank you for your feedback regarding the detail required in the caption for Figure 16. We have updated the figure and its caption to provide a clearer and more comprehensive description of the depicted content.

The revised Figure 16 now includes the following detailed descriptions for each panel:

- Panel (a) illustrates the simulated results of sediment bed level changes at the estuary of Danchangcun, showcasing the deposition patterns within the area.

- Panel (c) presents the simulated bed level changes near the river island in Danchangcun, highlighting the cyclical deposition trends.

- Panel (e) depicts the simulated bed level changes at the front end of the sand mouth in Danchangcun, indicating active sediment scouring and deposition.

- Panel (g) examines the simulated bed level changes at the sand mouth in Danchangcun,

with two distinct locations demonstrating similar sedimentation trends.

Furthermore, Panels (b), (d), (f), and (h) now display the temporal variation of bed thickness at representative points, providing a detailed look at how sediment deposition evolves over time in these specific locations.

The revised caption for Figure 16 reads:

*"Figure 11: Simulated results of bed level changes and sediment deposition in Danchangcun. (a) displays the bed level changes at the estuary; (c) near the river island; (e) at the front end of the sand mouth; (g) at the sand mouth itself; (b), (d), (f), and (h) represent the temporal changes in bed thickness at selected points."*

**Point 32: Line 417 Figure 16h and not 14**

Response: Thank you for your careful review and for pointing out the reference error at line 417. We have corrected the figure reference from "Figure 14" to "Figure 16h" as you indicated. This error was indeed a typographical oversight, and we appreciate your diligence in ensuring the accuracy of our manuscript.

The revised sentence now reads:

*"Finally, Figure 11h examines sediment deposition …"*

**Point 33: Line 425 I think you mean under the influence of the neap tide.**

Response (Line 413): Thank you for your attentive review and for pointing out the need for clarification at line 425. We have made the necessary correction to accurately reflect the tidal influence. The text now correctly states that the flow direction changes under the influence of

the neap tide, not the spring neap tide, as previously mentioned.

The revised sentence reads:

*"April 27th, under the influence of the neap tide, ..."*

**Point 34: Line 426 Saying that slower currents led to enhanced sedimentation is enough. You don't need that sentence.**

Response: Thank you for your feedback on line 426. We have removed the sentence as suggested, streamlining our discussion to focus on the primary relationship between slower currents and enhanced sediment deposition.

**Point 35 (Conclusion): The same comment for the abstract applies here. The readers will want to know the conclusions and key messages and may have not read the entire manuscript to know where these specific locations are. In addition, as mentioned in a previous comment, there is an inconsistency about A,B and C which to this point, the authors always referred to as estuaries but here they refer to them as channels. In any case, a potential reader may have not read the full manuscript so it is useless to mention them as A,B and C. I would recommend to include more general statements in the conclusions so that the value and significance of the results can be emphasized and also how these contribute to the research on this topic.**

Response (Lines 550-565): Thank you for your feedback on our manuscript, particularly regarding the conclusion section. We have expanded and refined the conclusion to provide a more comprehensive summary of our findings and their implications. The revised conclusion

now includes a detailed discussion of the sediment transport dynamics, the role of wave action, and the significance of residual currents in the lower reaches of the Changhua River. It also emphasizes the broader implications of our research for coastal management and environmental planning.

The revised conclusion reads:

*"In conclusion, our comprehensive study on the sediment transport dynamics in the lower reaches of the Changhua River, Hainan, has yielded valuable insights into the complex interplay between wave action, current flow, and sediment deposition. Through the application of an integrated wave-current coupled sediment transport model with variable grain properties, we have successfully simulated and analyzed the sediment behavior under the combined influence of waves and currents, particularly in multiple sub-estuaries with low sediment concentrations.*

*Our findings reveal significant sediment deposition in both the estuary and lower reaches of the river, which is primarily driven by the prevailing northeast-southwest tidal current direction and wave action. This has led to the formation of a two-way sand mouth, further narrowing the estuary and contributing to the substantial sediment accumulation at the mouth of the Changhua River. Furthermore, our research underscores the significance of residual currents in directing sediment movement and the dispersion of pollutant substances in the study area. The consistent flow of residual currents towards Beili Bay suggests that sediment in the lower reaches of the Changhua River is systematically transported in this direction, highlighting the importance of understanding these currents for coastal management and environmental planning.*

*Overall, our study contributes to the understanding of sediment transport processes in coastal environments and provides a robust framework for future research and management strategies*

*in similar estuarine systems. The detailed analysis of sediment deposition and the validation of our model against observed data confirm the reliability and applicability of our approach, offering valuable insights for coastal and environmental researches."*

**Editor's "Detailed Comments.":**

**Point 1: Lines 113-118. "V" in (7) needs definition here.**

Response (Line 117): Thank you for your feedback on lines 113-118, specifically regarding the need to define the variable "V" in equation (7).

We have added a definition for "V" in the vicinity of equation (7) to clarify that it represents the velocity of the flow. The revised text now reads:

*"…V is an average velocity of the fluid flow; …"*

**Point 2: Line 132. This is not clear – missing verb? Maybe "The model of sediment transport calculates the influence . ."?**

Response (Line 133): Thank you for your feedback on line 132 of our manuscript. We have revised the sentence to improve clarity and grammatical accuracy. The revised sentence now reads:

*"The model of sediment transport calculates the influence of the waves …"*

**Point 3: Lines 168-169. "median grain diameters (0-1φ)"; please explain "φ".**

Response (Lines 167-168): Thank you for your feedback on lines 168-169 regarding the clarification needed for the term "φ". We have added an explanation for "φ" in the text,

referencing the Udden-Wentworth scale, which is a logarithmic scale used to classify sediment grain sizes. The revised sentence now reads:

*"Here, φ represents the Udden-Wentworth scale, a logarithmic scale used to classify sediment grain sizes (Wentworth, 1922)."*

We have also added a reference to the original work by C.K. Wentworth, which introduced the scale, to provide historical context and academic backing for our use of the term.

**Point 4: Line 172. ". . (Table 2) . ."**

Response (Line 172): Thank you for your feedback on line 172 regarding the reference to Table 2. We have revised the sentence to ensure that the reference is clear and complete. The revised sentence now reads:

*"Through these data (Table 2), we can conclude that the majority of the areas are characterized by turbulent, ...."*

**Point 5: Table 2. Grain size cannot be negative. Please explain "-1φ – 0".**

Response: Thank you for your feedback on Table 2 and the query regarding the grain size range "-1φ – 0". We have added a note to clarify the Udden-Wentworth scale, which is a logarithmic scale used to classify sediment grain sizes. On this scale, a decrease in grain size corresponds to an increase in the φ value, with -1φ-0 corresponding to a grain size of 1-2 mm.

**Point 6: Line 184. ". . (Table 3) . ."Line 246. "(Table 4)."Line 250. "(Table 5)."**

Response (Lines 182, 245, 249): Thank you for your feedback on the referencing of tables in

our manuscript. We have reviewed and revised the referencing of Tables 3, 4, and 5 to ensure they adhere to the journal's formatting guidelines. The sentences now read:

*"According to the classification criteria of the sorting coefficients by Focke–Ward (Table 3), ..."*

*"The calibrated model parameters are presented in Table 4."*

*"The parameters used in the model setups are based on the values listed in Table 5."*

**Point 7: Line 252. 36 or 40 directional sectors?**

Response (Line 250): Thank you for your attention to detail regarding the number of directional sectors mentioned at line 252. Upon reviewing our data and methods, we have confirmed that the wave model at the open boundary is defined by the JONSWAP spectrum with a spectral resolution of 40 frequency bins and 36 directional sectors. We have updated the text to reflect this accurate number.

The revised sentence now reads:

*"The wave model at the open boundary is defined by the JONSWAP spectrum, with a spectral resolution of 40 frequency bins and 36 directional sectors."*

**Point 8: Figure 10. You need to state the depth (or height above bottom) of the model current shown, and also the depth (or height above bottom) of the ADCP bin.**

Response (Line 298): Thank you for your feedback on Figure 10 and for highlighting the need to include depth information.

We have reviewed the depth settings for our model and ADCP measurements. The model currents are represented at a depth of 2 meters, and the ADCP measurements, as detailed in

Table 6, are taken at 20.9 meters and 22.8 meters above the bottom. We have updated the caption for Figure 10 to include this information:

*"Figure 6: Average current speed and direction verification across the water column. Current speed and direction verification. (a) speed verification of ADCP 01; (b) speed verification of ADCP 02; (c) verification of current direction of ADCP 01; (d) verification of current direction of ADCP 02"*

**Point 9: Line 305. ". . Figure 11b and 11c depict . . . Figure 11b shows . ."**

Response (Lines 303-304): Thank you for bringing the inconsistencies between the results analysis and the figures mentioned in lines 303-307 to our attention.

We have corrected the references in lines 303-307 from Figure 10b and c to Figure 7, as it aligns with the content discussed in the text.

The revised text now reads:

*"The hydrodynamic simulation outcomes, as depicted in Fig. 7, .... Figure 11b and 11c depict the flow field outside the estuary of the Changhua River. Figure 7b shows …"*

**Point 10: Line 316. ". . River. Figure 12a shows . ."**

Response (Line 303): Thank you for your careful review and for pointing out the reference error at line 316. We have corrected the figure reference from "Figure 11" to "Figure 7" as you indicated. This error was indeed a typographical oversight, and we appreciate your diligence in ensuring the accuracy of our manuscript.

The revised sentence now reads:

*"Figure 7a shows the flow field at 23:00 on April 23, 2023, …"*

**Point 11: Line 322. ". . Figure 12b . ."**

Response (Line 309): Thank you for your careful review and for pointing out the reference error at line 322. We have corrected the figure reference from "Figure 11" to "Figure 12" as you indicated. This error was indeed a typographical oversight, and we appreciate your diligence in ensuring the accuracy of our manuscript.

The revised sentence now reads:

*"Figure 7b shows the flow field at 13:30 on April 24, 2023, …"*

**Point 12: Line 376. ". . Fig. 14 . ."**

Response (Line 362): Thank you for your careful review and for pointing out the reference error at line 376. We have corrected the figure reference from "Figure 13" to "Figure 14" as you indicated. This error was indeed a typographical oversight, and we appreciate your diligence in ensuring the accuracy of our manuscript.

The revised sentence now reads:

*"To further analyze the simulation validation, Fig. 9 presents…"*

**Point 13: Line 402. I think you mean ". . and take this point to represent the whole area. . ."**

Response (Lines 387-388): Thank you for your feedback on line 402 and for raising the concern about the representation of the entire area. We have rephrased the sentence to focus

on the sediment deposition processes at a specific point within the study area, without implying that this point is representative of the whole region. The revised sentence now reads:

*"To elucidate the sedimentary characteristics of the study area, we extract the bed level change data of a particular point in the obvious change area of river bed."*

**Point 14: Line 417. "Finally, Figure 16h . ."**

Response (Line 403): Thank you for your careful review and for pointing out the reference error at line 417. We have corrected the figure reference from "Figure 14" to "Figure 11" as you indicated. This error was indeed a typographical oversight, and we appreciate your diligence in ensuring the accuracy of our manuscript.

The revised sentence now reads:

*"Finally, Figure 11h examines sediment deposition at the sand mouth, with two distinct locations showing similar sedimentation trends, ..."*

**Point 15: Line 425. Avoid "spring neap" which makes no sense. Better ". . influence of decreased tidal amplitude . ."**

Response (Line 413): Thank you for your feedback on line 425 regarding the use of the term "spring neap." We have revised the sentence to avoid the term "spring neap" and instead focus on the influence of decreased tidal amplitude on sediment deposition. The revised sentence now reads:

*"After April 27th, under the influence of neap tide, the reduced tidal range and slower currents led to enhanced sediment deposition."*

**Point 16: Line 426. Better ". .sizes, including finer particles."**

Response: Thank you for your feedback on line 426, along with the comments from the reviewers. After careful consideration of all the input received, we have decided to remove the sentence at line 426 from the manuscript. This decision was made to streamline the narrative and ensure that the discussion remains focused on the key findings of our study.

**Point 17: Line 552. "upward" –> "northward"?**

Response (Line 539): Thank you for your suggestion regarding the term "upward" at line 552.

Upon reviewing the context, we agree that "northward" is a more appropriate term to describe the direction in question. We have replaced "upward" with "northward" to accurately reflect the direction of the Eulerian residual current.

The revised sentence now reads:

*"The northward Eulerian residual current, upon encountering the sea area outside Changhua Port, ..."*

*Once again, we appreciate the time and effort you have dedicated to evaluating our manuscript. Your expertise and guidance have been invaluable in strengthening our research!*